# Tube Loss: A Novel Loss Function for Prediction Interval Estimation

**Pritam Anand**                                            *pritam_anand@dau.ac.in*
*Dhirubhai Ambani University (Formerly DA-IICT), Gandhinagar.*

**Tathagata Bandyopadhyay**                                 *tathagata_b@dau.ac.in*
*Dhirubhai Ambani University (Formerly DA-IICT), Gandhinagar.*

**Suresh Chandra**                                          *sureshiitdelhi@gmail.com*
*Ex-faculty, Indian Institute of Technology, Delhi.*

**Reviewed on OpenReview:** *https://openreview.net/forum?id=3vwPza62Rr&referrer*

## Abstract

This paper proposes a novel loss function called Tube Loss, developed for the simultaneous estimation of the lower and upper bounds of a Prediction Interval (PI) in regression problems, including probabilistic forecasting in autoregressive framework. The PIs obtained through empirical risk minimization using Tube Loss exhibit superior performance compared to those derived from existing approaches. A theoretical analysis confirms that the estimated PIs asymptotically attain a user-specified confidence level $1 - \alpha$.

A distinctive feature of Tube Loss is its ability to shift the PI along the support of the response distribution through a tunable parameter, allowing the intervals to better align with high-density regions of the distribution. This is especially valuable for generating tighter intervals when the response distribution is skewed. Moreover, the method allows further narrowing of PIs through recalibration.

Unlike several prior techniques, the empirical risk associated with Tube Loss can be efficiently optimized via gradient descent. Extensive experiments [1] demonstrate the robustness and accuracy of the proposed method in delivering high-quality PIs across a range of models, including kernel machines, neural networks, and probabilistic forecasting frameworks .

## 1 Introduction

In a regression framework, machine learning (ML) models aim to predict the value of a real-valued random variable $y$, commonly referred to as the dependent variable, given the value of a vector of input variables $\boldsymbol{x}$, known as the independent variable. However, providing only a point prediction of $y$ without an uncertainty quantification may not be useful for applications. **Uncertainty quantification (UQ)** becomes particularly critical when the stakes of prediction errors are high. Consider, for instance, the scheduling of a replacement for a vital but, expensive component in a nuclear reactor, where failure could be catastrophic. Merely reporting that the expected remaining lifespan is 2.5 years offers limited utility for planning preventive maintenance. On the other hand, stating that the component's lifespan is expected to lie between 2 and 3 years with 99% confidence conveys significantly more actionable information. Such ranges for the predicted values of $y$ are termed *prediction intervals (PIs)*, and are associated with a specified confidence level (e.g., 99% in this case).

Suppose we are given $n$ training samples $\{(\boldsymbol{x}_i, y_i) : i = 1, 2, \ldots, n\}$, and we must now predict the unknown value of $y_{n+1}$ at a test point $\mathbf{x} = \mathbf{x}_{n+1}$. We assume $\{(\boldsymbol{x}_i, y_i) : i = 1, 2, \ldots, n + 1\}$ are i.i.d. from an

---

[1]The code is available at https://github.com/ltpritamanand/Tube_loss

arbitrary joint distribution $p(\boldsymbol{x}, y)$ of the random vector $(\boldsymbol{x}, y)$, with feature vector $\boldsymbol{x} \in \mathcal{R}^p$ typically being a vector-valued input, and response variable $y \in \mathcal{R}$. For the sake of notational convenience, we use $(\boldsymbol{x}, y)$ to denote both the random vector as well as their specific values, whenever the distinction is evident from the context.

Given an algorithm $\mathcal{A}$ and a training set $\{(\boldsymbol{x}_i, y_i) : i = 1, 2, \ldots, n\}$, a regression model is fitted to the training set to generate $\hat{\mu}(\mathbf{x})$, the predicted value of $y$ at $\mathbf{x}$, obtained by minimizing the empirical risk

$$arg \min_{\mu} \sum_{i=1}^{n} \rho(y_i, \mu(\boldsymbol{x}_i)), \tag{1}$$

where, $\rho(y, \mu(\boldsymbol{x}))$ is a suitably chosen loss function representing deviation of $\mu(\mathbf{x})$ from $y$.

In conventional regression set up, typically $\hat{\mu}(\boldsymbol{x})$ is an estimate of the conditional expectation $\mathbb{E}(y \mid \boldsymbol{x})$ and $\rho(y, \mu(\boldsymbol{x})) = (y - \mu(\boldsymbol{x}))^2$. Often, a quantile of the conditional distribution $y|\mathbf{x}$, specifically the median instead of the mean, may be preferred as the predictor of $y$, especially when the conditional distribution is heavy-tailed.

In general, for quantifying the uncertainty associated with a predictor $\hat{\mu}(\boldsymbol{x})$ of $y_{n+1}$, a widely used measure is the *prediction interval* $[\hat{\mu}_1(\boldsymbol{x}_{n+1}), \hat{\mu}_2(\boldsymbol{x}_{n+1})]$ with a specified confidence level $1 - \alpha$, which is defined as

$$P\big[\hat{\mu}_1(\mathbf{x}_{n+1}) \leq y_{n+1} \leq \hat{\mu}_2(\mathbf{x}_{n+1})\big] \geq 1 - \alpha. \tag{2}$$

Equation (2) is formulated for any joint distribution $p(\boldsymbol{x}, y)$ and for any sample size $n$. The probability involved in this expression is *marginal*, taken over all samples $\{(\boldsymbol{x}_i, y_i) : i = 1, 2, \ldots, n+1\}$. A procedure aimed at constructing high-quality (**HQ**) PIs must satisfy two essential criteria.

First, it should achieve the desired confidence level $1 - \alpha$ without imposing strong distributional assumptions. Second, the resulting intervals should be as narrow as possible across the input space, thereby ensuring that the predictions remain maximally informative.

The quality of a prediction interval is typically evaluated using two metrics: the **Prediction Interval Coverage Probability (PICP)** and the **Mean Prediction Interval Width (MPIW)**, both computed on the test set $\{(\boldsymbol{x}_i, y_i) : i = n+1, n+2, \ldots, n+m\}$. These measures are formally defined as

$$PICP = \frac{1}{m} \sum_{i=n+1}^{n+m} k_i, \tag{3}$$

where,

$$k_i = \begin{cases} 1, & \text{if } y_i \in [\hat{\mu}_1(\mathbf{x}_i), \hat{\mu}_2(\mathbf{x}_i)], \\ 0, & \text{Otherwise,} \end{cases} \tag{4}$$

and

$$MPIW = \frac{1}{m} \sum_{i=n+1}^{n+m} [\hat{\mu}_2(\mathbf{x}_i) - \hat{\mu}_1(\mathbf{x}_i)], \tag{5}$$

respectively. A **HQ PI** is expected to minimize the **MPIW** while ensuring that the **PICP** remains greater than or equal to $1 - \alpha$, the desired confidence level.

To address uncertainty quantification in regression tasks, various methods for constructing PIs have been proposed in the literature. Traditional approaches such as the *delta method* (Hwang & Ding (1997); Chryssolouris et al. (1996)), the *Mean-Variance Estimation (MVE)* technique (Nix & Weigend (1994)), and *Bayesian inference*-based models (MacKay (1992)) typically assume that the observational noise follows an independent Gaussian distribution. However, the performance of these approaches tends to deteriorate significantly when the underlying data distribution is highly skewed or exhibits heavy tails.

A more direct and model-agnostic strategy for constructing a **PI** with confidence level of $1 - \alpha$ is to estimate $[F_q(\boldsymbol{x}), F_{q+1-\alpha}(\boldsymbol{x})]$ where, $F_q(\boldsymbol{x})$ is the $q \in (0, 1)$-th quantile of the conditional distribution of $y$ given $\boldsymbol{x}$

defined by $P(y \leq F_q(\boldsymbol{x})|\boldsymbol{x}) = q$. In quantile regression, an estimate of $F_q(\boldsymbol{x})$, say, $\hat{F}_q(\boldsymbol{x})$, is obtained by replacing $\rho(y, \mu(\boldsymbol{x}))$ in (1) with the pinball loss function (Koenker & Bassett Jr (1978), Koenker & Hallock (2001)), given by

$$\rho_q(y, \mu(\boldsymbol{x})) = \begin{cases} q(y - \mu(\boldsymbol{x})), & \text{if} \quad y \geq \mu(\boldsymbol{x}), \\ (q-1)(y - \mu(\boldsymbol{x})), & \text{otherwise.} \end{cases} \tag{6}$$

It is noteworthy that obtaining the lower and upper bounds of this interval involves solving two distinct optimization problems to estimate $\hat{F}_q(\boldsymbol{x})$ and $\hat{F}_{q+1-\alpha}(\boldsymbol{x})$. Furthermore, to ensure that the resulting interval qualifies as a **HQ PI**, one must select an optimal value of $q$ (within the feasible range satisfying $q+1-\alpha \in (0, 1)$) that minimizes the **MPIW** on the test set:

$$\text{MPIW} = \frac{1}{m} \sum_{i=n+1}^{n+m} \left[ \hat{F}_{q+1-\alpha}(\boldsymbol{x}_i) - \hat{F}_q(\boldsymbol{x}_i) \right].$$

Determining the optimal value of $q$ requires repeatedly solving two separate optimization problems over a grid of candidate $q$ values, rendering the overall procedure for estimating the **PI** computationally demanding.

A few questions arise naturally at this point: Given $q$, can the lower and upper bounds be estimated *simultaneously* solving a single optimization problem? In other words, for a given value of $q$, is it possible to obtain the **PI** or both the quantiles as a direct output of a single optimization process? Furthermore, if such a formulation is feasible, the quality of the resulting interval would evidently depend on the choice of the underlying loss function. Hence, the central question becomes: what kind of loss function can yield **good-quality PIs**?

To address these issues, this paper introduces a novel loss function that produces **good-quality PIs**, referred to as the *Tube Loss* function. It may be viewed as a two-dimensional generalization of the *pinball loss* function, endowed with distinctive properties that enable the simultaneous estimation of both the lower and upper bounds of a **PI**. The intervals derived using the Tube Loss are shown, both theoretically and empirically, to achieve the nominal coverage level asymptotically. A formal proof of this result is provided.

Moreover, by tuning a hyperparameter of the loss function, the resulting interval can be shifted along the support of the response distribution $y$, thereby better capturing regions of high data density. As demonstrated in the experimental analysis, this feature proves particularly effective in reducing the **MPIW**, especially when the response distribution is asymmetric. Additionally, further reduction in **MPIW** on the test set can be achieved by incorporating a penalty term in the loss function with a user-defined parameter, particularly when the **PICP** on the validation set substantially exceeds the nominal coverage level $1 - \alpha$.

In the experimental results presented in Section 4, the proposed Tube Loss is employed to generate **PIs** across various regression frameworks, including *Kernel Machines*, *Neural Networks*, Deep Neural Networks for *probabilistic forecasting* tasks. Apart from this, we also extend the **Tube loss** methodology for obtaining the conformal prediction sets and quantifying the uncertainty into the STS task estimation along with appropriate baseline comparisons. The **PIs** derived from the **Tube loss** are observed to outperform those obtained using existing methods.

The rest of the paper is structured as follows. Section 2 provides a review of existing approaches of **UQ** in regression framework. Section 3 presents the proposed **Tube loss** function, describing its interesting properties and advantages. Section 4 discusses the results of extensive numerical evaluations. Finally, Section 5 outlines the possible future research directions.

## 2 Related Work

In this section, we briefly review existing **UQ** methods for regression within **NN** frameworks. For effective decision support, **UQ** methods must aggregate both data (*aleatoric*) and model (*epistemic*) uncertainty into the final estimate. Data uncertainty arises from the intrinsic stochasticity of the conditional distribution $y \mid \mathbf{x}$. In contrast, model uncertainty stems from the instability of **NN** training, where different parameter

realizations can lead to varying predictions under identical settings. The **PI** estimation methods are typically designed to capture data uncertainty well.

## 2.1 Aleatoric Uncertainty through Prediction Interval Estimation

Based on their underlying assumptions, **PI** estimation method in **NN** framework can be categorized into distribution-based and distribution-free **PI** approaches. Within the distribution-based category, *Mean–Variance Estimation (MVE)* (Nix & Weigend (1994)) and *Mixture Density Networks (MDN)* (Bishop (1994)) are among the most widely used approaches. The former models the conditional distribution $y \mid \mathbf{x}$ as Gaussian, while the latter, a mixture of Gaussian distributions, respectively. A systematic survey of **PI** estimation methods developed within **NN** frameworks, with a particular emphasis on distribution-based **PI** approaches, is presented in (Khosravi et al. (2011b)).

Recent **NN** based approaches typically estimate **PI**s in a distribution-free setting and have been shown to produce better **PI** quality, particularly under heterogeneous and unknown noise conditions. One of the most well-established **PI** estimation approaches is through the Quantile Regression Neural Network (**QRNN**), which trains two independent **NNs** to estimate the lower and upper quantile bounds of the PI, namely $[F_q(\boldsymbol{x}), F_{q+1-\alpha}(\boldsymbol{x})]$, by minimizing the pinball loss with a suitable choice of the quantile level $q$. However, an alternative line of research has evolved independently of the traditional quantile regression framework, in which **PI**s are generated directly as solutions to a single optimization problem, bypassing explicit quantile estimation. This simultaneous estimation of the **PI bounds** often enables explicit minimization of the **PI** width within the training objective, leading to narrower **PI**s in most applications without compromising the target calibration.

### 2.1.1 Lower Upper Bound Estimation

In a distribution-free setting, Khosravi et al. (2011a) were the first to introduce a method for jointly estimating the lower and upper bounds of the **PI** by directly minimizing empirical risk via a loss function, denoted by $\mathcal{L}_{LB}$. We call this method as Lower Upper Bound Estimation (**LUBE**). For given training set $\{(\boldsymbol{x}_i, y_i) : i = 1, 2, ..., n\}$, the **LUBE** (Khosravi et al. (2011a)) trains a single **NN** that considers the simultaneous estimation of PI bound functions $[\hat{\mu}_1(\boldsymbol{x}), \hat{\mu}_2(\boldsymbol{x})]$ by minimizing the following problem

$$\min_{(\mu_1, \mu_2)} \sum_{i=1}^{n} \mathcal{L}_{LB}(y_i, \mu_1(\boldsymbol{x}_i), \mu_2(\boldsymbol{x}_i)) := \min_{(\mu_1, \mu_2)} \frac{1}{nR} \sum_{i=1}^{n} (\mu_1(\boldsymbol{x}_i) - \mu_2(\boldsymbol{x}_i))\big(1 + \gamma\,(\mathbf{PICP})e^{-\eta(\mathbf{PICP}-(1-\alpha))}\big), \quad (7)$$

where **PICP** is computed for the **PI** $[\mu_1(\boldsymbol{x}), \mu_2(\boldsymbol{x})]$ over training samples $\{(\boldsymbol{x}_i, y_i) : i = 1, 2, ..., n\}$. and $R = \max(y_i) - \min(y_i)$ denotes the range of the response values $y_i$. The parameter $\eta$ is a user-defined constant that amplifies small deviations between **PICP** and $1 - \alpha$. The indicator function $\gamma(\mathbf{PICP})$ is defined as 0 if $\mathbf{PICP} \geq (1-\alpha)$, and 1 otherwise.

**LUBE** has been extensively used by the researchers in various **NN**-based application settings, such as energy load predictions (Pinson and Kariniotakis, 2013; Quan et al., 2014), wind speed forecasting (Wang et al., 2017; Ak et al.,2013), prediction of landslide displacement (Lian et al., 2016), gas flow (Sun et al., 2017), and solar energy (Galvan et al., 2017).

A significant drawback of **LUBE** loss function is that it's a function of **PICP**, a step function, and its gradient is zero almost everywhere (with respect to Lebesgue measure). Thus, the gradient descent (**GD**) method could not be used to minimize empirical risk. This is inconvenient considering that **GD** is now the standard method for training NNs. Instead, they propose simulated annealing (SA), a non-gradient-based optimization method. Later, various other non-gradient-based methods are used, including Genetic Algorithms (AK et al., 2013), Gravitational Search Algorithms (Lian et al., 2016), PSO (Galvan et al., 2017), and Artificial Bee Colony Algorithms (Shen et al, 2018).

### 2.1.2 Quality-Driven Loss

Building on the work of Khosravi et al. (2011a), Pearce et al. (2018) propose the Quality-Driven (**QD**) loss function, and seeks the solution of the following optimization problem:

$$\min_{(\mu_1, \mu_2)} \sum_{i=1}^{n} \mathcal{L}_{QD}(y_i, \mu_1(x_i), \mu_2(x_i)) := \min_{(\mu_1, \mu_2)} \frac{\sum_{i=1}^{n}(\mu_2(x_i) - \mu_1(x_i)).k_i}{\sum_{i=1}^{n} k_i} + \lambda \frac{n}{\alpha(1-\alpha)} max(0, (1-\alpha) - PICP)^2, \quad (8)$$

where **PICP** is computed for the **PI** $[\mu_1(x), \mu_2(x)]$ over training sample $\{(x_i, y_i) : i = 1, 2, ..., n\}$ and $k_i$ is defined as

$$k_i = \begin{cases} 1, & \text{if } y_i \in [\mu_1(x_i), \mu_2(x_i)]. \\ 0, & \text{Otherwise.} \end{cases}$$

Also, $\lambda$ is the user-defined trade-off positive parameter As discussed in Pearce et al. (2018) (Section 3.3.1), like **LUBE**, **QD** loss also involves **PICP**, and thus, **GD** method cannot be used. To enable the use of **the GD** method, they propose approximating **PICP** as the product of two sigmoid functions, a technique previously used by Yan et al. (2004). They also suggest using an ensemble of **NN** models instead of a single **NN**. It is found to yield **PIs** with shorter average widths.

Further, Salem et al. (2020) extended the **QD** loss by incorporating two additional penalty terms: one penalizes the mean squared error to encourage accurate point predictions, while the other enforces the structural integrity of the prediction intervals by preventing crossings between the **PI** bounds. They termed this modified loss function as the Quality-Driven$^{+}$ (**QD**$^{+}$) loss, which has been empirically shown to be more effective in practice compared to **QD** loss for **PI** estimation tasks.

### 2.1.3 Simultaneous Quantile Regression (SQR)

Tagasovska and Lopez-Paz (2019) propose simultaneous estimation of a large number of quantiles by minimizing the empirical risk (cf. (1)) of standard quantile regression with a pinball loss function (cf. (6)). The minimization is carried out by using a stochastic gradient descent algorithm, sampling fresh random quantile levels $q \sim \mathcal{U}(0, 1)$ for each training sample and mini-batch during training. Thus, **SQR** provides an estimate of the entire conditional distribution of $y$ given $x$ and thus, it enables them to "model non-gaussian, skewed, asymmetric, multimodal, and heteroscedastic" noise.

Given all possible quantiles, to generate a **PI**, any pair that satisfies the coverage constraint may be chosen. Because it estimates a large number of quantiles rather than the two bounds of **PI**, it faces significantly more challenges in learning during training. In spite of its added flexibility, its performance in terms of attaining nominal coverage is found to be inferior to that of the standard interval regression methods, such as those based on **QD** loss function (cf. Pouplin et al. (2024)).

### 2.1.4 Relaxed Quantile Regression (RQR)

As an alternative to **SQR**, Pouplin et al. (2024) propose a novel loss function that directly outputs the bounds of **PI** with a target coverage $1 - \alpha$ without requiring the intermediate step of estimating quantiles. This approach is called Relaxed Quantile Regression (**RQR**). The **RQR** loss function (Pouplin et al. (2024)) is given by

$$\mathcal{L}_{1-\alpha}^{RQR}(u_1, u_2) = \begin{cases} (1-\alpha)u_1 u_2, & \text{if } u_1 u_2 \geq 0, \\ -\alpha u_1 u_2, & \text{if } u_1 u_2 < 0, \end{cases} \quad (9)$$

where, $u_1$ and $u_2$ are the errors, $y - \mu_1(x)$ and $y - \mu_2(x)$, respectively, representing the deviation of the $y$ value from the PI bounds . Notice that, for $u_1 u_2 < 0$, $y$ will be within **PI** $[\mu_1(x), \mu_2(x)]$ and for $u_1 u_2 \geq 0$, it is on or outside **PI**, but $\mathcal{L}_{1-\alpha}^{RQR}(u_1, u_2)$ remains positive in both cases. However, for smaller values of $\alpha$ (such as 0.01, 0.05, 0.1), the value of $\mathcal{L}_{1-\alpha}^{RQR}(u_1, u_2)$ is much smaller for $u_1 u_2 < 0$ than for $u_1 u_2 \geq 0$. The multipliers of $u_1 u_2$ are so chosen as to satisfy the coverage constraint. Figure 1 shows the plot of the **RQR** loss for

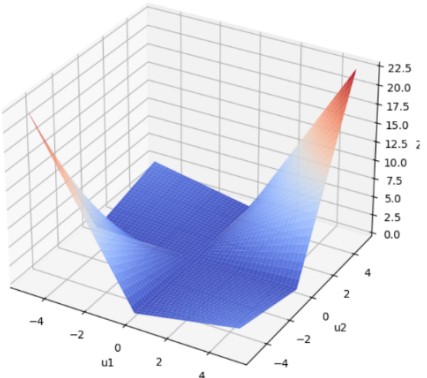

Figure 1: **RQR** loss function (Pouplin et al. (2024)) for 1-$\alpha$ =0.9

$1 - \alpha = 0.9$. Certainly, it is a nonconvex function of $u_1$ and $u_2$. For the training set $\{(\boldsymbol{x}_i, y_i) : i = 1, 2, \ldots, n\}$, **RQR** loss-based PI estimation model requires the solution of following optimization problem

$$\min_{\mu_1,\mu_2} \sum_{i=1}^{n} L_{1-\alpha}^{RQR}(y_i, \mu_1(x_i), \mu_2(x_i)) := \min_{\mu_1,\mu_2} \sum_{i=1}^{n} \begin{cases} 1 - \alpha(y - \mu_1(x))(y - \mu_2(x)), & \text{if } (y - \mu_1(x))(y - \mu_2(x)) \geq 0 \\ -\alpha(y - \mu_1(x))(y - \mu_2(x)), & \text{if } (y - \mu_1(x))(y - \mu_2(x)) < 0 \end{cases} \quad (10)$$

Pouplin et al. (2024) show that PI estimated through the **RQR** methodology enjoys several desirable theoretical properties, including asymptotic coverage, as well as unbiased and consistent estimation.

However, a major limitation of the **RQR** estimator is its high sensitivity to the presence of outliers. This sensitivity arises because, similar to expectile-based losses (Newey & Powell (1987)), the **RQR** loss is formulated as the product of two error terms representing the deviations of the true value from the lower and upper bounds of the **PI**. Consequently, outliers are more likely to induce crossings between the PI bounds, which can adversely affect the conditional coverage of the **PI**, particularly in regions near such crossings. Our experimental results in Section 4 substantiate this observation.

## 2.2 Epistemic Uncertainty in Prediction Interval model

Even when a **PI** estimation model yields high-quality intervals, the estimates themselves may still be subject to substantial model uncertainty. Such uncertainty can arise from sensitivity to training-set sampling, hyperparameter choices, and initialization conditions, leading to variability in the resulting PI estimates across different training runs. Therefore, effectively aggregating model and data uncertainties is essential to characterize the overall uncertainty associated with PI estimation accurately.

Bayesian methods (MacKay (1992)) quantify model uncertainty by placing prior distributions over model parameters and inferring their posterior distributions given the data, thereby propagating parameter uncertainty into predictive uncertainty. However, in the context of **NN** training, its estimates are sensitive to approximate inference and prior specification, and are often computationally expensive to implement, limiting their practicality for scalable uncertainty estimation.

Gal & Ghahramani (2016) developed Monte Carlo Dropout (**MC-dropout**) as a practical approach for estimating model uncertainty by activating dropout layers at test time. In this approach, dropout masks are randomly sampled and applied to the network weights for each forward pass, resulting in different sub-networks being activated at every evaluation. Repeating this process multiple times produces a collection of stochastic predictions, whose variability reflects the underlying model uncertainty.

Lakshminarayanan et al. (2017) developed a simple yet highly effective approach for quantifying model uncertainty in **NN** and deep learning models through **deep ensembles**. The central idea is to train the same **NN** architecture independently $s$ times under the *same hyperparameter setting*, while using different random

initializations of the model parameters $\boldsymbol{\theta}_i$, $i = 1, 2, \ldots, s$. Each trained model produces its own prediction, and the variability across these ensemble members is used to quantify model uncertainty.

Further, Pearce et al. (2018) aggregate the data uncertainty captured by the **NN**-based **PIs** with the model uncertainty arising from parameter variability using **deep ensembles** to obtain the final uncertainty estimate. Specifically, the **NN** is trained multiple times on the same training dataset with fixed hyperparameter settings but different random parameter initializations $\boldsymbol{\theta}_i$, producing a collection of **PIs** $\left[\mu_1^i(\mathbf{x}), \mu_2^i(\mathbf{x})\right]$, for $i = 1, 2, \ldots, s$. The aggregated **Ensemble PI** is then constructed by averaging the predicted lower and upper bounds across ensemble members and expanding them according to their empirical dispersion. This dispersion term captures model uncertainty, while the individual **NN** predictions account for data uncertainty, thereby yielding a unified uncertainty estimate for a given target coverage $1 - \alpha$.

## 3 Interval Estimation Using Tube Loss Function

In this section, we present the **Tube loss** function, which offers several advantages over the loss functions discussed earlier. First, unlike **RQR** loss, the **Tube loss** is a *linear* function of the errors, making the estimated bounds less sensitive to outliers. Second, in contrast to the intervals obtained through **RQR**, the bounds of the **Tube loss**-based interval never intersect in practice. Third, unlike the loss functions employed in the **QD** and **QD+** approaches, the **Tube loss** allows direct optimization of the empirical risk using the **GD** method. In **QD** and $\mathbf{QD}^+$ loss based PI models, a smooth function must approximate the loss function well to enable gradient-based optimization, and the quality of the resulting intervals depends heavily on the quality of such approximation. The experimental results presented in Section 4 vindicate this claim.

Fourth, a distinctive feature of the **Tube loss** is that its output intervals can be shifted along the support of $y$ by tuning a hyperparameter $r$. This flexibility allows the interval to better capture dense regions of the response distribution, thereby helping produce **tight** intervals, particularly when the conditional distribution of $y$ given $\mathbf{x}$ is asymmetric. Fifth, by introducing a user-defined regularization parameter $\delta$ into the minimization problem, the **Tube loss** enables an effective trade-off between **PICP** and **MPIW**, leading to narrower intervals in practice, especially when the **PICP** obtained on the validation set significantly exceeds the nominal coverage level $1 - \alpha$. Finally, we demonstrate that the intervals derived from the **Tube loss** function asymptotically attain the nominal coverage. In other words, as the training sample size $n \to \infty$, the empirical coverage converges to the nominal confidence level. A theoretical proof of this result is provided in Lemma 1 (cf. Section 3.2).

### 3.1 Tube Loss Function

Let us denote the prediction errors $(y - \mu_1)$ and $(y - \mu_2)$ by $u_1$ and $u_2$, respectively. For a given nominal coverage $(1 - \alpha)$ and $u_2 \leq u_1$ (i,e., $\mu_2 \geq \mu_1$), we define the **Tube loss** function as

$$
\mathcal{L}_{(1-\alpha)}^r(u_1, u_2) = \begin{cases} (1-\alpha)u_2, & \text{if } u_2 > 0, \\ -\alpha u_2, & \text{if } u_2 \leq 0, u_1 \geq 0 \text{ and } ru_2 + (1-r)u_1 \geq 0, \\ \alpha u_1, & \text{if } u_2 \leq 0, u_1 \geq 0 \text{ and } ru_2 + (1-r)u_1 < 0, \\ -(1-\alpha)u_1, & \text{if } u_1 < 0, \end{cases} \tag{11}
$$

where, $0 < r < 1$ is a user-defined parameter. In the following, we discuss a few properties of the **Tube loss** function.

First, in Figure 2, plots of the **Tube loss** are given for the default choice $r = 0.5$. Evidently, $\mathcal{L}_{(1-\alpha)}^{0.5}(u_1, u_2)$ is a symmetric continuous function around the line $u_1 + u_2 = 0$. Additionally, it is differentiable almost everywhere (with respect to the Lebesgue measure) and has non-zero gradients, thus enabling the direct application of the **GD** method.

The default choice of $r$ outputs centered **PIs**. However, by adjusting the value of $r$, the **PI** bounds can be shifted up or down, allowing the interval to capture the denser regions of the data cloud while maintaining coverage. It helps in reducing **MPIW** significantly, especially when the conditional distribution of $y$ given $x$ is skewed. The experimental results presented in Section 4 support this claim.

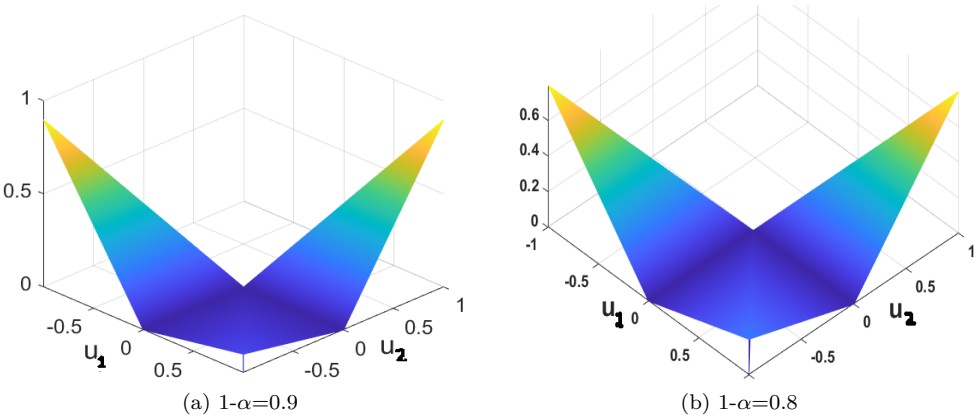

(a) 1-$\alpha$=0.9            (b) 1-$\alpha$=0.8

Figure 2: Tube loss function

### 3.2 Asymptotic properties of the Tube loss function

Let $y_1, y_2, ...., y_n$ be iid following the distribution of a continuous random variable $Y$ and $\alpha \in (0,1)$. Let us define the following disjoint subsets of $\mathbb{R}$: $\Re_1(\mu_1, \mu_2) = \{y : y > \mu_2\}$, $\Re_2(\mu_1, \mu_2) = \{y : \mu_1 < y < \mu_2, y > r\mu_2 + (1-r)\mu_1\}$, $\Re_3(\mu_1, \mu_2) = \{y : \mu_1 < y < \mu_2, y < r\mu_2 + (1-r)\mu_1\}$ and $\Re_4(\mu_1, \mu_2) = \{y : y < \mu_1\}$. Notice that the sets $\Re_i(\mu_1, \mu_2), i = 1, 2, 3, 4$ are so defined that they do not include the points on the boundaries. Suppose $(\mu_1^*, \mu_2^*)$ is the minimizer of $\frac{1}{n}\sum_{i=1}^{n} \mathcal{L}_{1-\alpha}^r(y_i - \mu_1, y_i - \mu_2)$ with respect to $(\mu_1, \mu_2)$, and $n_k$ denotes the number of data points in $\Re_k(\mu_1^*, \mu_2^*)$ ($k = 1, 2, 3, 4$). Then we have the following lemma.

**Lemma 1** *As $n \to \infty$, with probability 1 the following results hold: (i) $\frac{n_1}{n_2} \to \frac{\alpha}{1-\alpha}$, (ii) $\frac{n_4}{n_3} \to \frac{\alpha}{1-\alpha}$, and (iii) $\frac{n_1+n_4}{n_2+n_3} \to \frac{\alpha}{1-\alpha}$.*

The proof of the lemma is given in Appendix A.

We now consider the regression setup with training set $\boldsymbol{T} = \{(\boldsymbol{x}_i, y_i), i = 1, 2, \ldots, n\}$ where $(\boldsymbol{x}_i, y_i)$, $i = 1, 2, \ldots, n$ are iid following a density $p(\boldsymbol{x}, y)$ . Let us rewrite the **Tube loss** function (11) as

$$
\mathcal{L}_{(1-\alpha)}^r(y - \mu_1(\boldsymbol{x}), y - \mu_2(\boldsymbol{x})) = \begin{cases} (1-\alpha)(y - \mu_2(\boldsymbol{x})), & \text{if } y > \mu_2(\boldsymbol{x}), \\ -\alpha(y - \mu_2(\boldsymbol{x})), & \text{if } y \leq \mu_2(\boldsymbol{x}), y \geq \mu_1(\boldsymbol{x}) \text{ and } r(y - \mu_2(\boldsymbol{x})) + (1-r)(y - \mu_1(\boldsymbol{x})) \geq 0, \\ \alpha(y - \mu_1(\boldsymbol{x})), & \text{if } y \leq \mu_2(\boldsymbol{x}), y \geq \mu_1(\boldsymbol{x}), \text{ and } r(y - \mu_2(\boldsymbol{x})) + (1-r)(y - \mu_1(\boldsymbol{x})) < 0, \\ -(1-\alpha)(y - \mu_1(\boldsymbol{x})), & \text{if } y < \mu_1(\boldsymbol{x}), \end{cases}
$$

that can be simplified as

$$
\mathcal{L}_{(1-\alpha)}^r(y, \mu_1(\boldsymbol{x}), \mu_2(\boldsymbol{x})) = \begin{cases} (1-\alpha)(y - \mu_2(\boldsymbol{x})), & \text{if } y > \mu_2(\boldsymbol{x}). \\ -\alpha(y - \mu_2(\boldsymbol{x})), & \text{if } \mu_1(\boldsymbol{x}) \leq y \leq \mu_2(\boldsymbol{x}) \text{ and } y \geq r\mu_2(\boldsymbol{x}) + (1-r)\mu_1(\boldsymbol{x}). \\ \alpha(y - \mu_1(\boldsymbol{x})), & \text{if } \mu_1(\boldsymbol{x}) \leq y \leq \mu_2(\boldsymbol{x}) \text{ and } y < r\mu_2(\boldsymbol{x}) + (1-r)\mu_1(\boldsymbol{x}). \\ -(1-\alpha)(y - \mu_1(\boldsymbol{x})), & \text{if } y < \mu_1(\boldsymbol{x}). \end{cases} \tag{12}
$$

For the training set $\boldsymbol{T}$, let $(\hat{\mu}_1^r(\boldsymbol{x}), \hat{\mu}_2^r(\boldsymbol{x}))$ be the minimizer of the empirical risk $\frac{1}{n}\sum_{i=1}^{n} \mathcal{L}_{1-\alpha}^r(y_i, \mu_1(\boldsymbol{x}_i), \mu_2(\boldsymbol{x}_i))$, where $\mu_1(\boldsymbol{x})$ and $\mu_2(\boldsymbol{x})$ belong to an appropriately chosen class of functions that is sufficiently regular and measurable so that uniform convergence of empirical risk to population risk holds. With some abuse of notation, we let $n_k^r$ denote the cardinality of the set $\Re_k(= \Re_k(\hat{\mu}_1^r(\boldsymbol{x}), \hat{\mu}_2^r(\boldsymbol{x})))$ ($k = 1, 2, 3, 4$) inducing a partition of the feature space (as shown in Figure 3). We then have the following proposition.

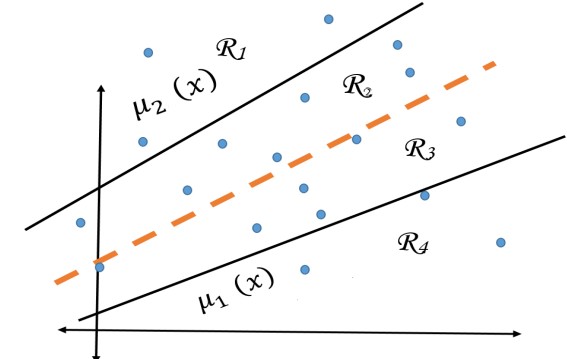

Figure 3: The **Tube loss** induces a partition of the feature space into four regions. Red line represents the convex combination of $\mu_1(\boldsymbol{x})$ and $\mu_2(\boldsymbol{x})$, i,e., $r\mu_2(\boldsymbol{x}) + (1-r)\mu_1(\boldsymbol{x})$, $0 < r < 1$. Blue dots represent data points $(\boldsymbol{x}_i, y_i)$, $i = 1, 2, ,.., n$.

**Proposition 1** *For $\alpha \in (0,1)$ as $n \to \infty$ the following results hold with probability $1$,*
*(i) $\frac{n_1^r}{n_2^r} \to \frac{\alpha}{1-\alpha}$, (ii) $\frac{n_4^r}{n_3^r} \to \frac{\alpha}{1-\alpha}$ (iii) $\frac{n_1^r + n_4^r}{n_2^r + n_3^r} \to \frac{\alpha}{1-\alpha}$, provided $(\boldsymbol{x}_i, y_i), i = 1, 2, ..., n$ are iid following a distribution $p(\boldsymbol{x}, y)$ with $p(y|\boldsymbol{x})$ continuous and the expectation of the modulus of absolute continuity of its density [1] satisfies $\lim_{\delta \to 0} E[\epsilon(\delta)] = 0$.*

Proof: The proof follows from the proof of the Lemma stated above and Lemma 3 of (Takeuchi et al. (2006)).

### 3.3 Parameter $r$ and movement of the PI bounds

Now, we show that how the PI bounds can be moved up and down by choosing $r$ appropriately. Consider two values of $r$, say $r_1$ and $r_2$ such that $0 < r_1 < r_2 < 1$. Let us denote the cardinality of the set $\Re_k(\hat{\mu}_1^{r_i}(\mathbf{x}), \hat{\mu}_2^{r_i}(\mathbf{x}))$ by $n_k^{r_i}$ ($k = 1, 2, 3, 4$, $i = 1, 2$.)

Now, Proposition 1 (iii) entails that asymptotically $1 - \alpha$ fraction of $y$ values should lie inside each of the intervals $[\hat{\mu}_1^{r_1}(\boldsymbol{x}), \hat{\mu}_2^{r_1}(\boldsymbol{x})]$ and $[\hat{\mu}_1^{r_2}(\boldsymbol{x}), \hat{\mu}_2^{r_2}(\boldsymbol{x})]$.[2] But, $r_1 < r_2$ simply entails $\frac{n_2^{r_1}}{n_3^{r_1}} > \frac{n_2^{r_2}}{n_3^{r_2}}$, which in turn implies $\frac{n_1^{r_1}}{n_4^{r_1}} > \frac{n_1^{r_2}}{n_4^{r_2}}$ from (i) and (ii) of Proposition 1 for large $n$. Thus, changing the value of $r$ from $r_1$ to $r_2$ moves the **PI** tube up and vice versa.

The optimal choice of $r$ in the **Tube loss** depends on the characteristics of the conditional distribution $y \mid \mathbf{x}$. When $r = 0.5$, the **Tube loss** implicitly assumes that $y \mid \mathbf{x}$ is symmetrically distributed, resulting in a **centered PI** estimate with the narrowest achievable width. In this setting, the **PI** is calibrated such that, asymptotically, a fraction $\alpha/2$ of the observations lies in regions $\Re_1$ and $\Re_4$, respectively. Moreover, the mid function $r\,\mu_2(\boldsymbol{x}) + (1-r)\,\mu_1(\boldsymbol{x})$ (illustrated by the orange line in Figure 3) coincides with the median of the conditional distribution $y \mid \boldsymbol{x}$.

For asymmetric noise distributions, the optimal value of $r$ is closely related to the skewness of the noise distribution. In the presence of right-skewed noise, smaller values of $r$ shift the PI downward, enabling the construction of a narrower interval. Conversely, for left-skewed noise distributions, larger values of $r$ are more appropriate and lead to better quality PI.

In Section 4, we demonstrate that, in several experiments, selecting an appropriate value of $r$ results in a significant reduction of **MPIW**. Notice that, $\mathcal{L}_{1-\alpha}^r(u_1, u_2)$ is discontinuous at the separating plane

---

[1]The modulus of absolute continuity of a function f is defined as the function $\epsilon(\delta) = Sup \sum_i |f(b_i) - f(a_i)|$, where the supremum is taken over all disjoint intervals $(a_i, b_i)$ with $a_i < b_i$ satisfying $\sum_i (b_i - a_i) < \delta$. Loosely speaking, the conditional density $p(y|x)$ is absolutely continuous on average. This ensures that the probability of a point lying on the boundaries of PI vanishes.

[2]The *asymptotic calibration* guarantees established in Proposition 1 do not necessarily lead to *asymptotic individual calibration* guarantees (Chung et al. (2021)), that strictly requires $P\big(\mu_1(x) \le y \le \mu_2(x) \mid x\big) = 1 - \alpha$.

$\{(u_1, u_2) : ru_2 + (1-r)u_1 = 0\}$ when $r \neq 0.5$. However, while running the experiments, we observe that it has not caused any problem for the implementation of the GD method.

## 3.4 Recalibrating Prediction Intervals to Balance MPIW and PICP

After choosing an appropriate value of $r$, if the attained coverage (**PICP**) of the output interval in the validation set exceeds the nominal coverage $1 - \alpha$, then we go for re-calibration, which helps in further minimization of MPIW, and thus, enhances the quality of the interval, keeping the attained coverage more than equal to $1 - \alpha$. For re-calibration, we employ the following loss function,

$$\mathcal{L}_{\mathcal{W}(1-\alpha)}^r(y, \mu_1(\boldsymbol{x}), \mu_2(\boldsymbol{x})) = \mathcal{L}_{(1-\alpha)}^r(y, \mu_1(\boldsymbol{x}), \mu_2(\boldsymbol{x})) + \delta|\mu_2(\boldsymbol{x}) - \mu_1(\boldsymbol{x})|. \tag{13}$$

The default choice of $\delta$ is zero, unless there is room for recalibration, i.e, if **PICP** is more than $1 - \alpha$ in the validation data set. Notice that in contrast to RQR, the penalty term in our case uses a linear function of the errors, thus making it less sensitive to outliers. In Section 4, we employ recalibration in some experiments to enhance the quality of the **PI**.

## 4 Experiments

In this section, we present the experimental results on generating **PIs** in kernel machines, neural networks, deep learning for probabilistic forecasting task and analyze the **Tube loss** methodology in comparison with existing losses in the literature. Apart from this, we extend the **Tube loss PI** model for obtaining the conformal prediction set and quantifying the uncertainty in semantic textual similarity task along with the appropriate baseline comparisons.

In general, to compare two **PI** estimation methods, say $\mathcal{M}_1$ and $\mathcal{M}_2$, we adopt the following rule in this paper: method $\mathcal{M}_1$ is considered superior to method $\mathcal{M}_2$ if either **PICP**$(\mathcal{M}_1) \geq 1 - \alpha$ and **PICP**$(\mathcal{M}_2) < 1 - \alpha$, or if $\min\{$**PICP**$(\mathcal{M}_1),$ **PICP**$(\mathcal{M}_2)\} \geq 1 - \alpha$ and **MPIW**$(\mathcal{M}_1) <$ **MPIW**$(\mathcal{M}_2)$.

Furthermore, in experiments conducted with synthetic data, where samples are generated from a known distribution, the true conditional quantiles of $y$ given $\boldsymbol{x}$ are known for all $\boldsymbol{x}$. Consequently, the corresponding true **PI**, denoted by $[\mu_1(\boldsymbol{x}), \mu_2(\boldsymbol{x})]$, is available. In such settings, the accuracy of an estimated PI, $[\hat{\mu}_1(\boldsymbol{x}), \hat{\mu}_2(\boldsymbol{x})]$, is assessed using the *Sum of Mean Squared Errors (**SMSE**)*, on the test set defined as

$$\frac{1}{m} \sum_{i=1}^m \left(\hat{\mu}_1(\boldsymbol{x}_i) - \mu_1(\boldsymbol{x}_i)\right)^2 + \frac{1}{m} \sum_{i=1}^m \left(\hat{\mu}_2(\boldsymbol{x}_i) - \mu_2(\boldsymbol{x}_i)\right)^2.$$

### 4.1 Tube loss in kernel machines

In kernel machines, the **PI** bounds are a pair of functions

$$\mu_1(\boldsymbol{x}) := \sum_{i=1}^n k(\boldsymbol{x}_i, \boldsymbol{x})\eta_i + \eta_0 \quad \text{and} \quad \mu_2(\boldsymbol{x}) := \sum_{i=1}^n k(\boldsymbol{x}_i, \boldsymbol{x})\beta_i + \beta_0, \tag{14}$$

where $k(\boldsymbol{x}, y)$ is positive definite kernel (Mercer (1909)). Using matrix notation, we rewrite $\mu_1(\boldsymbol{x})$ and $\mu_2(\boldsymbol{x})$ as

$$\mu_1(\boldsymbol{x}) := K(A^T, \boldsymbol{x})\boldsymbol{\eta} + \eta_0 \quad \text{and} \quad \mu_2(\boldsymbol{x}) := K(A^T, \boldsymbol{x})\boldsymbol{\beta} + \beta_0, \tag{15}$$

where $A$ is an $n \times p$ data matrix containing $n$ training points in $\mathbb{R}^p$, $\boldsymbol{\eta} = \begin{bmatrix} \eta_1 \\ \eta_2 \\ .. \\ \eta_n \end{bmatrix}$, $\boldsymbol{\beta} = \begin{bmatrix} \beta_1 \\ \beta_2 \\ .. \\ \beta_n \end{bmatrix}$ and $K(A^T, \mathbf{x}) = $

$\left[k(\boldsymbol{x}_1, \boldsymbol{x}), k(\boldsymbol{x}_2, \boldsymbol{x}), .., k(\boldsymbol{x}_n, \boldsymbol{x})\right]$. The Tube loss-based kernel machines consider the following optimization

problem for outputting the **PI**:

$$\min_{(\boldsymbol{\eta},\boldsymbol{\beta},\eta_0,\beta_0)} \frac{\lambda}{2}(\boldsymbol{\eta}^T\boldsymbol{\eta} + \boldsymbol{\beta}^T\boldsymbol{\beta}) + \sum_{i=1}^{n} \mathcal{L}^r_{(1-\alpha)}\big(y_i, \big(K(A^T,\mathbf{x}_i)\boldsymbol{\eta} + \eta_0\big), \big(K(A^T,\boldsymbol{x}_i)\boldsymbol{\beta} + \beta_0\big)\big)$$

$$+ \delta \sum_{i=1}^{n} \big|(K(A^T,\boldsymbol{x}_i)(\boldsymbol{\eta} - \boldsymbol{\beta}) + (\eta_0 - \beta_0)\big|. \tag{16}$$

The Gradient descent solution to the above **Tube loss** kernel optimization problem is derived in Appendix B.

For obtaining high-quality **PI**s, the parameters $r$ and $\delta$ must be appropriately tuned using the validation dataset. The default settings $r = 0.5$ and $\delta = 0$, typically yield good performance when the conditional distribution of $y$ given $\boldsymbol{x}$ is symmetric, since in that case a centered prediction interval is optimal. However, if the distribution is positively (negatively) skewed, selecting an $r$ value smaller (larger) than 0.5 shifts the bounds downward (upward), thereby minimizing **MPIW** while maintaining **PICP** at the desired level. Once an appropriate $r$ has been determined, if the resulting **PICP** exceeds the target confidence level $1 - \alpha$, recalibrating through $\delta$ can further reduce **MPIW** without compromising the intended coverage.

### 4.1.1 Experimental Results

**1. Choice of $r$**

We first show, in the context of kernel-based models, that the choice of the parameter $r$ is crucial for achieving narrow **PI** bounds, especially when the underlying data distribution exhibits asymmetry.

To illustrate this, we construct two synthetic datasets, denoted by $\mathbf{D_1}$ and $\mathbf{D_2}$. Each dataset consists of observations $\{(x_i, y_i) : i = 1, 2, \ldots, 1500\}$, where $x_i$ is sampled from the uniform distribution defined over the interval $[0, 1]$, and the response variable $y_i$ is generated according to

$$y_i = \frac{\sin x_i}{x_i} + \epsilon_i, \tag{17}$$

where $\epsilon_i$ represents a random noise component. For dataset $\mathbf{D_1}$, $\epsilon_i$ follows a symmetric normal distribution $\mathcal{N}(0, 0.8)$, while for dataset $\mathbf{D_2}$, $\epsilon_i$ follows a positively skewed distribution given by $\chi^2(3) - 3$.

The model is trained on 500 data points, with the remaining samples reserved for testing. For constructing linear PI tubes, a linear kernel is employed. For nonlinear PI estimation, we adopt the radial basis function (RBF) kernel defined as

$$k(\mathbf{x}_1, \mathbf{x}_2) = e^{-\gamma\|\mathbf{x}_1 - \mathbf{x}_2\|^2},$$

where $\gamma$ is a suitably chosen hyperparameter.

For dataset $\mathbf{D_1}$, the **Tube loss** function is applied with a linear kernel to construct **PI**s at nominal confidence levels of $1 - \alpha = 0.8$ and $1 - \alpha = 0.9$, using the default parameter settings $r = 0.5$ and $\delta = 0$. The obtained **PICP** and **MPIW** values for the test dataset corresponding to $1 - \alpha = 0.8$ (0.9) are 0.799 (0.909) and 2.10 (2.78), respectively. Figure 4 illustrates the estimated **PI**s on the test data for both target confidence levels, $1 - \alpha = 0.8$ and $1 - \alpha = 0.9$.

For dataset $\mathbf{D_2}$, we construct the **PI** corresponding to $1 - \alpha = 0.8$ using the RBF kernel. The **Tube loss**-based kernel machine with $r = 0.5$ and $\delta = 0$ produces a centered interval, as shown in Figure 5(b), yielding **PICP** and **MPIW** values of 0.79 and 6.47, respectively. Owing to the positively skewed nature of the data distribution, such a centered interval is expected to produce a relatively wider **PI** (i.e., higher **MPIW**) compared to a **HQ PI**. To effectively capture the denser regions of the data, the **PI** bounds therefore need to be shifted downward.

Figures 5(a), 5(c), and 5(d) depict the **PI**s obtained from the **Tube loss**-based kernel machines for $r = 0.8$, 0.3, and 0.1, respectively. It is evident that as $r$ decreases, the estimated **PI** bounds move downward, covering the denser portions of the data and thereby reducing the **MPIW**.

Figure 6 (a) presents the variation of **MPIW** with respect to $r$, clearly showing that smaller values of $r$ lead to lower **MPIW**, as expected. Figure 6(b) displays the plots of **PICP**, lower quantile (LQ), and upper

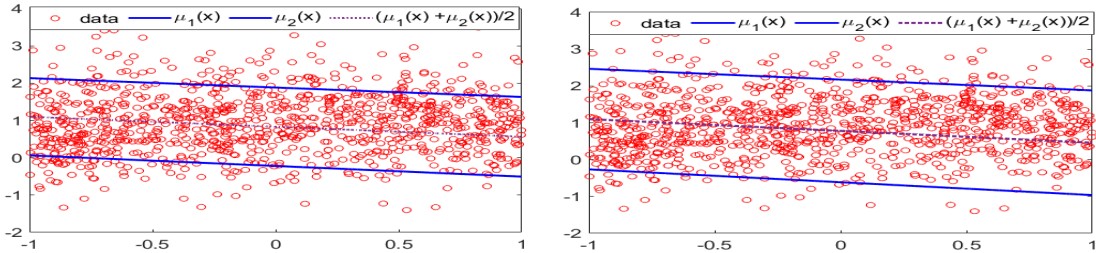

Figure 4: Tube loss-based PI estimation for $\mathbf{D_1}$ with (a)$1 - \alpha = 0.8$ and (b) $1 - \alpha = 0.9$.

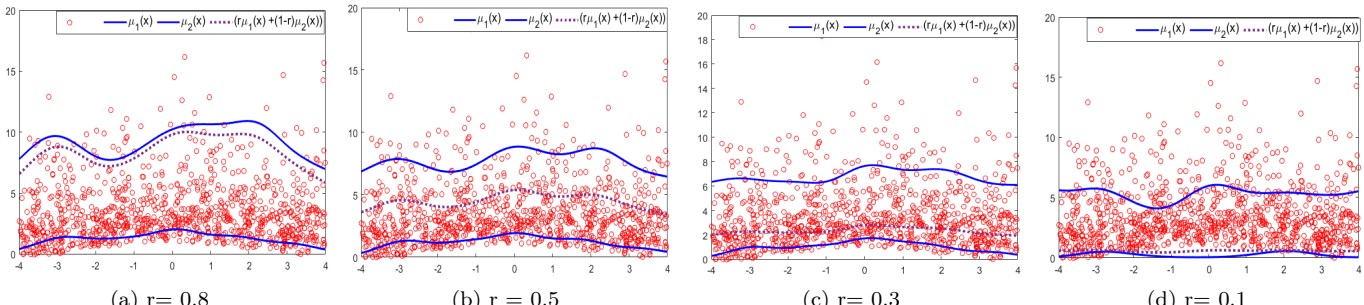

(a) r= 0.8       (b) r = 0.5       (c) r= 0.3       (d) r= 0.1

Figure 5: Tube loss-based PI estimation for $\mathbf{D_2}$ with $1 - \alpha = 0.8$ and $r = 0.1, 0.3, 0.5, 0.8$.

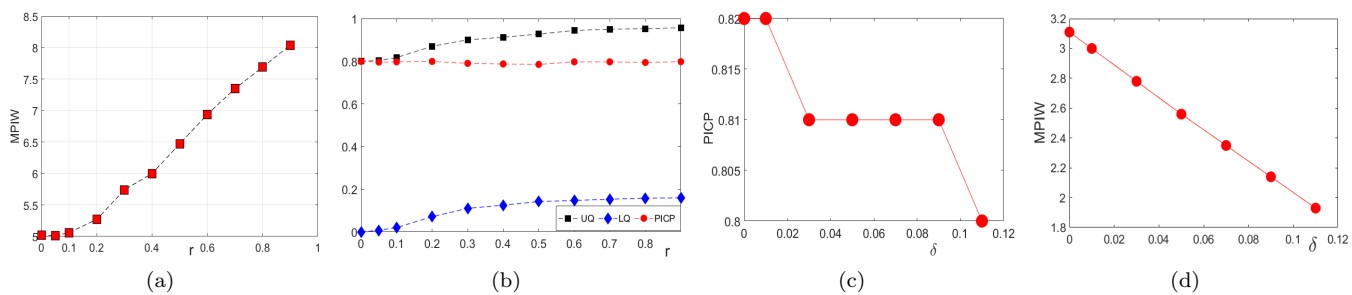

(a)       (b)       (c)       (d)

Figure 6: Plot of (a) r against MPIW ,(b) PCIP, UQ and LQ for dataset $\mathbf{B}$. Plot of (c) $\delta$ against PICP,(d) $\delta$ against MPIW on the Servo dataset.

|  | Model | (q+1-$\alpha$,q)/(r,$\delta$) | PICP | MPIW | Time (s) |
|---|---|---|---|---|---|
| $\mathbf{D_1}$ | Q Ker M | (0.95,0.15) | $0.78 \pm 0.017$ | $2.088 \pm 0.109$ | 219.38 |
|  | Q Ker M | (0.90,0.10) | $0.78 \pm 0.019$ | $2.002 \pm 0.068$ | 221.72 |
|  | Q Ker M | (0.85,0.05) | $0.79 \pm 0.025$ | $2.151 \pm 0.078$ | 217.38 |
|  | T Ker M | (0.6,0) | $0.80 \pm 0.012$ | $2.165 \pm 0.068$ | 56.93 |
|  | T Ker M | (0.5,0) | $0.80 \pm 0.018$ | $2.156 \pm 0.069$ | 61.50 |
| $\mathbf{D_2}$ | Q Ker M | (0.90,30) | $0.59 \pm 0.032$ | $4.6609 \pm 0.153$ | 138.05 |
|  | Q Kerl M | (0.95,0.35) | $0.58 \pm 0.023$ | $5.5369 \pm 0.272$ | 146.60 |
|  | Q Ker M | (0.80,20) | $0.59 \pm 0.027$ | $3.6196 \pm 0.143$ | 146.41 |
|  | T Ker M | (0.3,0) | $0.60 \pm 0.032$ | $3.495 \pm 0.168$ | 39.07 |
|  | T Ker M | (0.2,0) | $0.601 \pm 0.028$ | $3.174 \pm 0.165$ | 41.55 |

Table 1: Quantile-based Kernel Machine (Q ker M) and Tube loss-based kernel Machine (T ker M) on dataset $\mathbf{D_1}$ and $\mathbf{D_2}$. Q ker M involves parameters (q+1-$\alpha$,q) and T ker M involves (r, $\delta$).

quantile (UQ) against $r$, where LQ and UQ denote the proportions of $y_i$ values falling below the upper and lower bounds of the estimated PI tube, respectively. The plots indicate that changing $r$ affects both LQ and UQ but leaves **PICP** largely unchanged. Notably, reducing $r$ from 0.5 to 0.1 results in a significant improvement in **MPIW** by approximately 21.80%, without compromising the desired coverage.

It is important to note that for dataset $\mathbf{D_1}$, where the noise terms are drawn from a symmetric distribution, the default choice of $r = 0.5$ yields a **HQ PI**. In contrast, for dataset $\mathbf{D_2}$, characterized by a positively skewed distribution, obtaining a good quality **PI** requires reducing the value of $r$ from its default setting of 0.5.

Finally, we compare the performance of **PIs** obtained using the **Tube loss** with those based on the pinball loss, where the latter estimates the two bounds $\hat{F}_q(\boldsymbol{x})$ and $\hat{F}_{q+(1-\alpha)}(\boldsymbol{x})$ separately. We generate 10 independent replicates each of datasets $\mathbf{D_1}$ and $\mathbf{D_2}$. For every replicate, we compute the **PICP** and **MPIW** of the PIs derived from both methods at nominal confidence levels of $1 - \alpha = 0.8$ and 0.6, respectively.

It is worth noting that for quantile-based **PI** estimation, the choice of the parameter $q$, and for **Tube loss**-based estimation, the choice of the parameter $r$, are both critical for producing **HQ** intervals. Table 1 summarizes the comparative results. The Tube Loss-based method not only offers a significant reduction in overall training complexity, thereby decreasing training time, but also consistently achieves the desired nominal confidence level across both datasets.

For dataset $\mathbf{D_1}$, the centered interval corresponding to $q = 0.1$ under the pinball loss framework is expected to perform best. Although its **PICP** falls marginally short of the target value of 0.8, it yields a slightly narrower interval compared to the **Tube loss** -based PI with $r = 0.5$. As expected for symmetric noise distributions, the **Tube loss** with $(r = 0.5, \delta = 0)$ provides the most reliable **PI** for dataset $\mathbf{D_1}$.

For dataset $\mathbf{D_2}$, however, the **MPIW** values of the estimated **PIs** can be substantially improved by reducing the value of $r$, which aligns with expectations since the noise component follows a positively skewed distribution.

## 2. Choice of $\delta$

As discussed earlier, we now demonstrate how an appropriate choice of the recalibration parameter $\delta$ (cf. Equation (16)) can further reduce the **MPIW** on the test data. To illustrate this, we consider the well-known Servo dataset ( Karl Ulrich (2016)), which contains 167 observations across five variables. A randomly selected subset comprising 10% of the data points is used as the test set, while the remaining data are utilized for training with a 10-fold cross-validation procedure to obtain a Tube Loss-based **PI** (using the linear kernel) at a nominal confidence level of 0.8, with default settings of $r$ and $\delta$. The mean **PICP** observed on the validation set is 0.82. Since the empirical **PICP** exceeds the target level of 0.8, there exists an opportunity to reduce the **MPIW** on the test set by selecting a positive value of $\delta$. We incrementally increase $\delta$ until the validation **PICP** reaches the target level of 0.8, as depicted in Figure 6(c). This adjustment results in approximately a 44% reduction in **MPIW** on the test set, as shown in Figure 6 (d), while preserving the desired coverage.

### 4.2 Tube Loss in Neural Network (NN)

The **Tube loss** based PI estimation model within the **NN** framework aims to solve the following optimization problem using the training set $T = \{(\boldsymbol{x}_i, y_i) : i = 1, 2, \ldots, n\}$:

$$\min_{(\mu_1, \mu_2)} \left[ \sum_{i=1}^{n} \mathcal{L}_{(1-\alpha)}^{r} \big(y_i, \mu_1(\boldsymbol{x}_i), \mu_2(\boldsymbol{x}_i)\big) + \delta \sum_{i=1}^{n} \big|\mu_2(\boldsymbol{x}_i) - \mu_1(\boldsymbol{x}_i)\big| \right], \tag{18}$$

where $\mu_1(\boldsymbol{x})$ and $\mu_2(\boldsymbol{x})$ represent the two output neurons of the dense network corresponding to the lower and upper bounds of the **PI**, respectively. The weights of both the hidden and output layers are iteratively updated through gradient-based optimization using backpropagation of the loss function.

#### 4.2.1 Comparison with QD Loss

Within the **NN** framework, Pearce et al. (2018) demonstrated that **PIs** obtained using the **QD** loss yield slightly superior performance compared to those derived from the **LUBE** approach proposed by Khosravi et al. (2011a). Furthermore, they observed that the **QD** method also outperforms the mean-variance estimation (MVE) approach of Nix & Weigend (1994) on several benchmark datasets.

In this study, we conduct a simple experiment to show that the **Tube loss**-based **PI** provides better performance than the **QD**-based **PI** in terms of the *sum of mean squared errors (SMSE)*—a local measure of smoothness and tightness introduced earlier in Section 4. To this end, we generate 1000 data points $(x_i, y_i)$ according to the following model:

$$y_i = \frac{\sin(x_i)}{x_i} + \epsilon_i, \tag{19}$$

where $x_i \sim U(-2\pi, 2\pi)$ and $\epsilon_i \sim U(-1, 1)$.

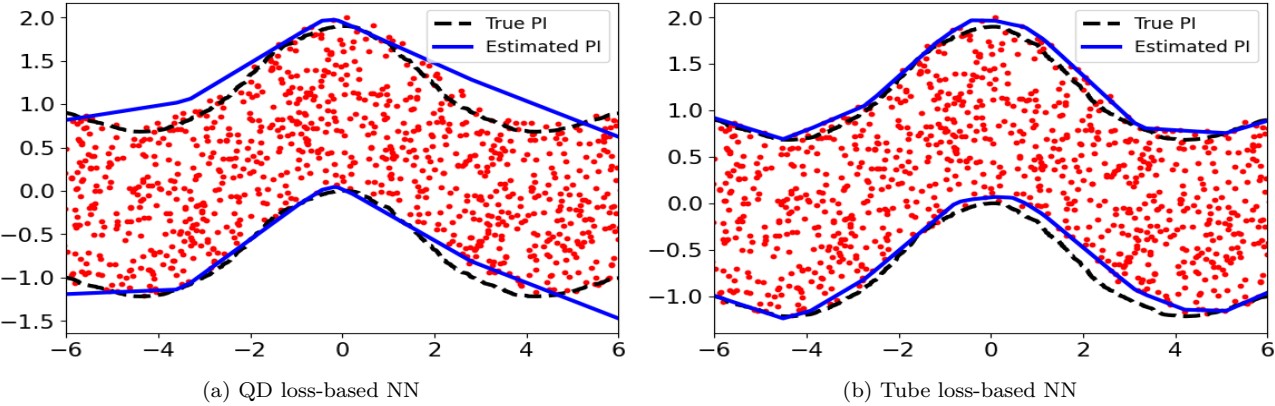

(a) QD loss-based NN

(b) Tube loss-based NN

Figure 7: Comparison between Tube Loss and QD Loss function-based neural networks.

Both the **Tube loss**-based and **QD loss**-based NN models are trained to obtain **PIs** at a nominal confidence level of 0.95. After hyperparameter tuning, the **NN** architecture is fixed with 100 ReLU-activated hidden neurons and two output neurons. The *Adam* optimizer is employed for model training. Figure 7(a) presents the **PI** estimated using the QD loss, while Figure 7(b) displays the **PI** obtained via the Tube Loss with $r = 0.5$ $\delta = 0$, alongside the true **PI**. It is evident that the **PI** generated by the **Tube loss** is smoother and more compact compared to that produced by the **QD** loss. The true **PI** is computed from the 97.5% and 2.5% quantiles of the conditional distribution of $y|x$ for various values of $x$.

The observed (**PICP**, **MPIW**) values for the **PIs** obtained using the **Tube loss** and **QD** loss are $(0.95, 1.78)$ and $(0.98, 2.17)$, respectively. For the **QD** loss, proper adjustment of the softening parameter $s$, which governs the approximation of the step function via the product of sigmoid functions, is crucial but often cumbersome. Through the tuning process, the optimal parameters for the **QD** loss were identified as $s = 200$ and $\lambda = 0.1$.

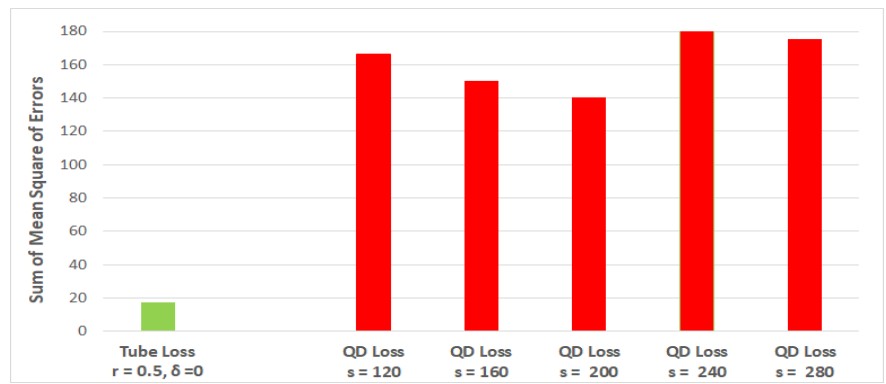

Figure 8: Tube Loss-based NN approximates the true PI more closely than the QD loss-based NN.

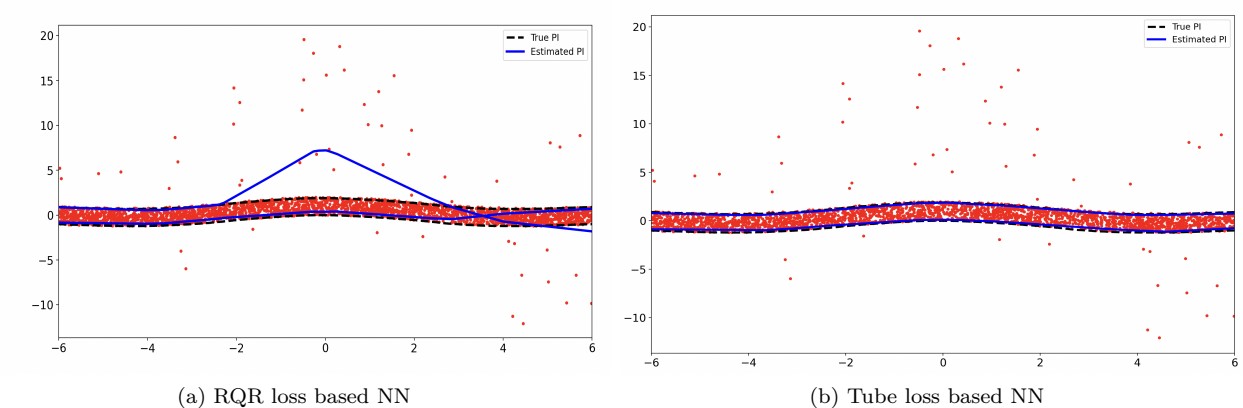

(a) RQR loss based NN            (b) Tube loss based NN

Figure 9: Comparison of RQR loss and Tube loss function based NNs

Subsequently, we computed the **SMSEs** of the resulting **PIs**. Figure 8 illustrates the comparison of **SMSE** values between the **Tube loss** and **QD** loss for various choices of the softening parameter $s$. It is evident that the performance of the **QD** loss-based **PI** is highly sensitive to the selection of $s$, and consequently, to the fidelity of the step function approximation achieved by the sigmoid functions.

### 4.2.2 Comparison with RQR loss

As stated before, the **RQR PI** bounds are sensitive to outliers. The presence of outliers may trigger the crossing of the two bounds, thus adversely affecting the conditional coverage of the **PI** near the crossing. We perform a simple experiment to show the impact of outliers on **RQR** based **PI** . It is worth mentioning in this context that for comparison with **Tube loss**, we consider the RQR-Width Minimizing (**RQR-W**) method. We generate an artificial data set $\{(x_i, y_i), i = 1, 2, ....3000\}$ using the relation.

$$y_i = \frac{sin(x_i)}{x_i} + \epsilon_i, \tag{20}$$

where $x_i$ is $U(-2\pi, 2\pi)$ and $\epsilon_i$ is from $U(-1, 1)$. Next, we contaminate the dataset by adding 60 (2%) data points $\{(x_i, 10y_i), i = 3001, 3002, ....3060\}$, where $\{(x_i, y_i), i = 3001, 3002, ....3060\}$ are generated randomly. The **Tube loss** and **RQR** Loss-based **NNs** were trained using a single-layer **NN** with 100 hidden neurons, ReLU activation, a batch size of 100, 1200 epochs, and the Adam optimizer for a target confidence level of 0.8. For both loss functions, the penalty parameters $\delta$ in **Tube Loss** and $\lambda$ in **RQR** were set to zero. Figure 9 presents the results of **RQR**- and **Tube Loss**–based **NNs** with $r = 0.3$. In 9(a), the **RQR**-based **PI** deviates considerably from the true **PI** (black dotted line), and the **PI** bounds cross each other near $x = 4$.

In contrast, the **Tube loss**-based PI almost overlaps with the true **PI**, effectively handling the asymmetric noise introduced by the outliers. This example clearly supports our apprehension stated above.

| Dataset | TUBE | RQR-W | QR | SQR-C | SQR-N | QD loss |
|---------|------|-------|-----|-------|-------|---------|
| **miami** | 89.34 (0.45) | 88.89 (0.32) | 90.50 (0.57) | 87.36 (0.46) | 85.97 (0.48) | 90.26 (0.49) |
| | 0.49 (0.02) | 0.51 (0.01) | **0.51** (0.01) | 1.90 (0.05) | 1.52 (0.03) | 1.74 (0.12) |
| **wine** | 91.21 (0.37) | 90.16 (0.37) | 89.97 (0.36) | 91.94 (0.67) | 88.19 (0.97) | 88.03 (0.92) |
| | **0.26** (0.017) | 0.33 (0.01) | 0.28 (0.00) | 0.49 (0.03) | 0.48 (0.03) | 1.18 (0.12) |
| **power** | 90.31 (0.54) | 89.34 (0.55) | 89.92 (0.54) | 91.81 (1.34) | 89.11 (1.40) | 62.47 (13.63) |
| | **0.04** (0.009) | 0.07 (0.01) | 0.06 (0.01) | 0.15 (0.03) | 0.15 (0.03) | 1.14 (0.23) |
| **yacht** | 89.32 (0.41) | 90.32 (0.42) | 91.45 (1.23) | 85.97 (1.18) | 84.68 (1.69) | 90.32 (0.96) |
| | **0.16** (0.019) | 0.33 (0.02) | 0.32 (0.02) | 3.56 (0.38) | 2.91 (0.26) | 1.66 (0.29) |
| **kin8nm** | 89.00 (0.32) | 89.27 (0.40) | 91.89 (0.27) | 87.16 (0.49) | 85.53 (0.53) | 89.52 (0.46) |
| | 0.42 (0.01) | 0.42 (0.01) | **0.50** (0.01) | 1.14 (0.01) | 1.10 (0.01) | 1.61 (0.12) |
| **protein** | 92.16 (0.29) | 88.47 (0.30) | 90.40 (0.23) | 89.01 (0.20) | 86.01 (0.32) | 90.38 (0.30) |
| | 1.68 (0.02) | 1.50 (0.01) | **1.61** (0.01) | 2.19 (0.02) | 2.05 (0.01) | 1.96 (0.09) |
| **boston** | 91.58 (0.48) | 88.14 (0.66) | 86.67 (1.06) | 82.55 (1.48) | 81.18 (1.58) | 90.29 (0.77) |
| | **0.44** (0.025) | 0.40 (0.02) | 0.38 (0.01) | 1.09 (0.03) | 1.01 (0.02) | 1.41 (0.15) |
| **energy** | 90.08 (0.42) | 89.68 (0.70) | 93.05 (0.88) | 89.42 (0.99) | 86.30 (0.99) | 89.42 (0.76) |
| | **0.23** (0.016) | 0.23 (0.01) | 0.29 (0.01) | 1.31 (0.01) | 1.23 (0.01) | 1.39 (0.17) |
| **sulfur** | 92.00 (0.36) | 88.87 (0.20) | 88.30 (0.34) | 87.83 (0.47) | 87.40 (0.50) | 89.41 (0.53) |
| | **1.06** (0.027) | 1.02 (0.01) | 1.05 (0.01) | 1.09 (0.03) | 1.02 (0.02) | 1.36 (0.07) |
| **cpu_act** | 91.03 (0.35) | 89.14 (0.37) | 88.94 (0.48) | 90.85 (0.75) | 86.73 (1.10) | 90.32 (0.62) |
| | **0.44** (0.014) | 0.44 (0.00) | 0.41 (0.01) | 0.78 (0.01) | 0.76 (0.02) | N/A* (N/A*) |
| **concrete** | 91.12 (0.52) | 88.74 (0.66) | 87.77 (1.09) | 86.02 (1.29) | 85.29 (1.39) | 89.37 (0.68) |
| | **0.46** (0.026) | 0.47 (0.03) | 0.44 (0.01) | 1.39 (0.03) | 1.33 (0.02) | 1.28 (0.16) |
| **naval** | 91.01 (0.34) | 88.81 (0.22) | 88.34 (0.25) | 90.70 (1.68) | 88.86 (1.77) | 91.16 (0.35) |
| | **0.02** (0.008) | 0.03 (0.00) | 0.02 (0.00) | 0.26 (0.04) | 0.26 (0.04) | 2.93 (0.18) |

Table 2: Comparisons of **Tube loss** based intervals against that of based on **RQR-W** and other existing baseline **NN** methods on **12** datasets for the **PI** estimation task with target $1 - \alpha = 0.90$. The first row reports the **PICP**, while the second row reports **MPIW**. All results represent the test set mean over 10 runs ($\pm$ standard error). Best results are in bold; the best **PI** method attains average confidence level at least 0.90 with the lowest **MPIW**. Tuned parameters are listed in Appendix C.

### 4.2.3 Comparison on benchmark datasets

Next, we assess the performance of the **Tube loss**-based **PI** in comparison with the distribution-free PI baseline approaches discussed in Section 2, using twelve benchmark datasets. In particular, we examine the PIs generated by **RQR-W** (Pouplin et al. (2024)), **QRNN** (Quantile Regression Neural Network), **SQR-C** (centered intervals) (Tagasovska & Lopez-Paz (2019)), **SQR-N** (potentially non-centered and typically the narrowest) (Tagasovska & Lopez-Paz (2019)), and **QD** (Pearce et al. (2018)). The numerical results summarized in Table 2 are produced under the experimental setup described in (Pouplin et al. (2024)). Details regarding the selection of the tuning parameters $r$ and $\delta$ are presented in Appendix C.

Among the twelve datasets considered, the **Tube loss**-based intervals exhibit superior performance in nine cases, whereas the QR-based intervals perform best in the remaining three. These results clearly demonstrate the advantage of **Tube loss**-based intervals over existing approaches in generating high-quality prediction intervals (**PIs**) within the **NN** framework.

### 4.3 Tube Loss based deep Probabilistic Forecasting

Further, we use the **Tube loss** function in Long Short Term Memory (**LSTM**) Network (Hochreiter & Schmidhuber (1997)) for obtaining the **probabilistic forecast** upon popular benchmark time-series datasets namely Electric (BP & Ember. (2016)), Sunspots (SIDC & Quandl.), SWH (NDBC),Temperature (machinelearningmastery.com) , Female Birth (datamarket.com) and Beer Production (Australian (1996)). Also, we compare the **Tube loss** based **LSTM** (T-LSTM) with **Quantile based LSTM** (Q-LSTM) on

these datasets. The 70% initial data points were considered for training and rest of them were test set. Out of training sets, last 10% of observations were used as validation set. For each dataset, we tuned the **LSTM**

| Dataset | PICP | | MPIW | | Training Time (SEC) | | Improvement | Better |
|---|---|---|---|---|---|---|---|---|
| | Q-LSTM | T- LSTM | Q-LSTM | T-LSTM | Q-LSTM | T-LSTM | in Time(s) | |
| Electric | 0.95 | 0.95 | 18.23 | **17.02** | 145 | 58 | 60 % | $\checkmark$ |
| Sunspots | 0.93 | **0.95** | 121.09 | 113.98 | 839 | 442 | 47 % | $\checkmark$ |
| SWH | 0.96 | 0.96 | 0.52 | **0.35** | 5119 | 2733 | 46 % | $\checkmark$ |
| Temperature | **0.95** | 0.94 | 24.82 | 15.56 | 1135 | 447 | 60 % | |
| Female Birth | 0.95 | 0.96 | 28.20 | **28.09** | 118 | 43 | 63 % | $\checkmark$ |
| Beer Production | 0.94 | **0.95** | 134.8 | 42.91 | 132.8 | 89.6 | 33 % | $\checkmark$ |

Table 3: Comparison of **Quantile based LSTM** (Q-LSTM) and **Tube loss based LSTM** (T-LSTM) on real-world time-series datasets with target coverage $1 - \alpha = 0.95$. Tuned parameters and plots obtained by the T-LSTM are listed in Appendix C.

architecture, dropout, and window size based on prior literature, then applied **Tube loss** and the quantile approach separately to target a 0.95 PI. Table 3 shows that the T-LSTM model obtains better performance on five cases out of 6 datasets. It also tends to obtain lower **MPIW** than Q-LSTM model, as the $\delta$ parameter facilitates the re-calibration for capturing the narrower **PI** in T-LSTM model. But, main advantages of T-LSTM over the Q-LSTM is the significant improvement in training time . It is because that the Q-LSTM needs to be trained twice where as T-LSTM requires only single cycle of training.

In a distribution-free setting, quantile-based deep learning models are the most popular and established choice for probabilistic forecasting. However, we also need to compare the performance of the **Tube loss** based probabilistic forecasting with that of the deep learning models trained with the **QD loss** function (Pearce et al. (2018)) . A significant practical challenge in training deep learning models with the **QD loss** function (Pearce et al. (2018)) is the frequent occurrence of NaN losses during computation. However, the extended Quality Driven (**QD$^+$**) loss function (Salem et al. (2020)) solves this problem up to some extent. Consequently, we have implemented the **QD$^+$** loss function based deep forecasting models for obtaining the probabilistic forecast. We tune the parameter values of **QD$^+$** loss function based **LSTM** well and compare its performance with the **Tube loss based LSTM** model on time-series benchmark dataset at Table 4. Unlike, the Quantile and **Tube loss based LSTM** models, the **QD$^+$** loss function based **LSTM** model fails to obtain the consistent performance. Out of six datasets, it fails to obtain the target coverage on three datasets. In case of Sunspots and Temperature datasets, the **QD$^+$** loss function based **LSTM** obtains very poor performance. Additionally, across all datasets, the **Tube loss LSTM** models outperform the **QD$^+$** loss **LSTM** models.

| Dataset | PICP | | MPIW | | Improvement | Better |
|---|---|---|---|---|---|---|
| | $QD^+$-LSTM | T- LSTM | $QD^+$-LSTM | T-LSTM | in MPIW | |
| Electric | 0.96 | 0.95 | 54.63 | **17.02** | 68.84 % | $\checkmark$ |
| Sunspots | 0.43 | **0.95** | 24.06 | 113.98 | NA | $\checkmark$ |
| SWH | 0.96 | 0.96 | 0.36 | **0.35** | 2.78% | $\checkmark$ |
| Female Birth | 0.94 | **0.96** | 38.98 | 28.09 | 27.94% | $\checkmark$ |
| Temperature | 0.79 | **0.95** | 5.94 | 15.56 | 73.13 % | $\checkmark$ |
| Beer Production | 0.96 | 0.95 | 159.71 | **42.91** | NA | $\checkmark$ |

Table 4: Comparison of Extended Quality Driven loss based LSTM (**QD$^+$ LSTM**) and **Tube loss based LSTM** (T-LSTM) on real-world time-series datasets with target coverage $1 - \alpha = 0.95$

For numerical results presented in the Table 3, the tuned parameters are detail in the Table 9. Also, the plot obtained by the Tube loss based LSTM for benchmark datasets reported in Table 3, are shown in the 11.

## 4.4   Application to wind probabilistic forecasting

We have also employed the **Tube loss** function in **LSTM**, **GRU** and **TCN** for **probabilistic forecasting of wind speed**. Table 5 lists the performance of the **Tube loss** based deep learning models along with

the recent benchmark models used for **probabilistic forecasting** in recent wind energy literature on San Francisco wind dataset containing 26,304 hourly observations.The last 30% of observations were used as the test set, while the final 10% of the training data was reserved for validation.

| Rank | Model | PICP | MPIW | MPIW/PICP |
|------|-------|------|------|-----------|
| 1 | **TCN + Tube** | **0.9543** | **4.560** | **4.78** |
| 2 | **LSTM + Tube** | **0.9561** | **4.627** | **4.84** |
| 3 | **GRU + Tube** | **0.9507** | **4.857** | **5.11** |
| 4 | GRU + Quantile | 0.951 | 4.955 | 5.21 |
| 5 | LSTM + Quantile | 0.9594 | 5.407 | 5.64 |
| 6 | TCN + QD$^+$ | 0.9505 | 5.490 | 5.78 |
| 7 | LSTM + QD$^+$ | 0.9734 | 6.043 | 6.21 |
| 8 | Deep AR | 0.9855 | 6.2023 | 6.29 |
| 9 | GRU + QD$^+$ | 0.9724 | 6.309 | 6.49 |
| 10 | MDN | 0.949 | 4.620 | 4.87 |
| 11 | TCN + Quantile | 0.9428 | 4.908 | 5.21 |
| 12 | TimeGPT | 0.9357 | 11.235 | 12.00 |

Table 5: Ranking of different deep probabilistic forecasting methods including Deep AR (Salinas et al. (2020)), Mixed Density Network (MDN) (Bishop (1994)), QD $^+$ based deep learning models (Salem et al. (2020)) and Time GPT (Garza et al. (2023)) on San Francisco Dataset with target coverage $1 - \alpha = 0.95$. DeepAR (Salinas et al. (2020)) has recently gained traction in probabilistic forecasting, including wind power prediction (Arora et al. (2022)), while studies (Yang et al. (2021), Zhang et al. (2020), Men et al. (2016)) highlight the effectiveness of Mixture Density Networks (MDN) for wind forecasting.

Table 5 shows that the **Tube loss**–based deep learning models consistently achieve the best overall performance among all compared baselines.

### 4.5 Conformal Prediction with Tube Loss

Here we present an empirical study to compare the performance of Tube Loss based **conformal PI** with that of based on quantile regression.

As introduced earlier, let $\boldsymbol{T} = \{(\boldsymbol{x}_i, y_i)\}_{i=1}^n$ denote a set of $n$ training samples drawn from a joint distribution $p(\boldsymbol{x}, y)$. The objective is to construct a **PI** for a future response $y_{n+1}$ of the form $[\hat{\mu}_1(\boldsymbol{x}_{n+1}), \hat{\mu}_2(\boldsymbol{x}_{n+1})]$ with nominal coverage $1 - \alpha$. Assuming exchangeability of the augmented dataset $\{(\boldsymbol{x}_1, y_1), (\boldsymbol{x}_2, y_2), \ldots, (\boldsymbol{x}_{n+1}, y_{n+1})\}$, Romano et al. (2019) proposed **Conformalized Quantile Regression (CQR)**, a quantile-regression–based **conformal framework**. By introducing quantile-based nonconformity scores, their method yields a fully adaptive conformal prediction set $C(\boldsymbol{x}_{n+1})$ for $y_{n+1}$ that provably achieves the target coverage level $1 - \alpha$ for any finite sample size.

Under the split **conformal regression (CR)** framework (cf. Papadopoulos et al. (2001); Papadopoulos (2008)), the original training set $T$ is first divided into two mutually exclusive subsets: $I_1$, which is used to fit the predictive model, and $I_2$, reserved for calibration. Subsequently, nonconformity scores are evaluated on each instance in the calibration set $I_2$. These scores quantify the deviation between the model's prediction $\hat{y}_i$ for input $x_i$ and the corresponding ground-truth response $y_i$.

Within the **NN** framework, the **CQR** approach begins by training two separate models on the training set $I_1$, each minimizing the pinball loss to estimate the $q$-th and $(1 + q - \alpha)$-th conditional quantile functions, respectively, where $0 \leq q$, $(1 + q - \alpha) \leq 1$. Let the resulting estimators be denoted by $\hat{F}_q(x)$ and $\hat{F}_{1+q-\alpha}(x)$. Using these estimates, the nonconformity scores $E_i$ are computed on the calibration set $I_2$ as

$$E_i = \max\left\{\hat{F}_q(x_i) - y_i, \ y_i - \hat{F}_{1+q-\alpha}(x_i)\right\}. \tag{21}$$

Subsequently, following the **conformal prediction** framework, the empirical $(1 - \alpha)\left(1 + \frac{1}{|I_2|}\right)$-quantile $Q_{1-\alpha}(E, I_2)$ of the set $\{E_i : i \in I_2\}$ is evaluated. The adaptive **PI** for a new test input $x_{n+1}$ is then

constructed as

$$C(x_{n+1}) = \left[ \hat{F}_q(x_{n+1}) - Q_{1-\alpha}(E, I_2), \ \hat{F}_{1+q-\alpha}(x_{n+1}) + Q_{1-\alpha}(E, I_2) \right]. \tag{22}$$

Similar to the **CQR** method discussed above, in **Tube loss** based **conformal PI** estimation, we replace the quantile bounds $\hat{F}_q(x)$ and $\hat{F}_{1+q-\alpha}(x)$ by the corresponding **Tube loss** based bounds $\hat{\mu}_1(\boldsymbol{x})$ and $\hat{\mu}_2(\boldsymbol{x})$, respectively in (35) and (36).

| Dataset | PICP | | Error | | MPIW | | Time (sec) | |
|---------|------|------|-------|------|------|------|------------|------|
| | CQR | TCQR | CQR | TCQR | CQR | TCQR | CQR | TCQR |
| Concrete | $0.90 \pm 0.04$ | $0.91 \pm 0.03$ | 2 | 3 | $17.11 \pm 1.76$ | $19.89 \pm 1.46$ | $34.04 \pm 2.66$ | $17.32 \pm 1.99$ |
| Bike | $0.90 \pm 0.01$ | $0.90 \pm 0.01$ | 3 | 3 | $192.51 \pm 43.23$ | $171.77 \pm 9.12$ | $74.36 \pm 7.05$ | $40.21 \pm 4.31$ |
| Star | $0.90 \pm 0.02$ | $0.90 \pm 0.02$ | 3 | 3 | $1022.19 \pm 40.99$ | $982.83 \pm 43.13$ | $32.06 \pm 2.36$ | $16.28 \pm 1.16$ |
| Community | $0.90 \pm 0.02$ | $0.91 \pm 0.02$ | 5 | 1 | $0.43 \pm 0.02$ | $0.47 \pm 0.02$ | $17.30 \pm 3.61$ | $9.01 \pm 0.18$ |
| Facebook | $0.91 \pm 0.01$ | $0.90 \pm 0.00$ | 0 | 1 | $18.51 \pm 4.46$ | $17.78 \pm 1.63$ | $182.81 \pm 5.48$ | $99.35 \pm 3.53$ |
| Boston Housing | $0.89 \pm 0.03$ | $0.92 \pm 0.03$ | 6 | 2 | $9.38 \pm 0.73$ | $13.12 \pm 3.70$ | $4.73 \pm 0.06$ | $2.66 \pm 0.07$ |
| Naval | $0.90 \pm 0.01$ | $0.90 \pm 0.01$ | 1 | 4 | $0.02 \pm 0.00$ | $0.02 \pm 0.00$ | $83.10 \pm 6.92$ | $46.70 \pm 4.28$ |
| Yatch Hydro | $0.86 \pm 0.04$ | $0.90 \pm 0.05$ | 7 | 4 | $2.37 \pm 0.71$ | $2.47 \pm 0.56$ | $41.72 \pm 8.54$ | $20.39 \pm 3.39$ |
| Auto MPG | $0.92 \pm 0.04$ | $0.93 \pm 0.04$ | 2 | 1 | $10.03 \pm 0.94$ | $9.90 \pm 1.24$ | $53.21 \pm 4.64$ | $28.11 \pm 4.18$ |

Table 6: Comparison of the Tube loss–based Conformal Regression (TCR) method with Conformalized Quantile Regression (CQR) (Romano et al. (2019)) across multiple benchmark datasets for a target coverage level of $1 - \alpha = 0.90$.

To benchmark the proposed **Tube-loss–based Conformal Regression (TCR)** against Conformalized Quantile Regression (**CQR**) (Romano et al. (2019)), we conduct experiments on nine widely used benchmark datasets (cf. Table 10). Each dataset is randomly partitioned into training, calibration, and test subsets following a 3:1:1 split. **PIs** are constructed at the target confidence level $1 - \alpha = 0.9$, and the entire procedure is repeated 10 times.

Table 6 reports the mean **PICP** and **MPIW** computed on the test sets. In addition, we present under the column labeled "**Error**", the number of trials out of the 10 repetitions in which a method fails to attain the desired coverage level of $1 - \alpha = 0.9$. As shown in Table 10, neither approach consistently outperforms the other in terms of **PICP** and **MPIW**, indicating broadly comparable performance, with the **TCR** exhibiting marginal improvements over **CQR** in several cases. Finally, with respect to computational efficiency, **TCR** clearly demonstrates superior runtime performance.

### 4.6 Application to Semantic Textual Similarity task

Given a sentence pair $(S_1, S_2)$, the **Semantic Textual Similarity (STS)** task predicts their semantic relatedness on a continuous scale $[0, k]$, where $k$ is a positive integer. As **STS** is inherently a subjective regression problem, it is affected by annotation noise and semantic ambiguity, making point predictions alone insufficient for reliable inference. To address this limitation, recent work advocates **PI** estimation as a principled form of uncertainty quantification, explicitly accounting for both data (aleatoric) and model (epistemic) uncertainty (Wang et al. (2022)). Within this framework, **PI**-based approaches aim to learn lower and upper bounds that enclose the true similarity score denoted with $w$. Such uncertainty-aware **STS predictions** enhance trustworthiness, improve robustness, and better support downstream decision-making in practical applications.

To facilitate a comparative assessment of the proposed **Tube loss NN** against existing **PI** estimation approaches on the **STS** task, we employ the widely used **STS-B** benchmark dataset Cer et al. (2017). For each **PI** model, model (epistemic) uncertainty is quantified using an **ensemble**-based strategy. The resulting model uncertainty is subsequently combined with the corresponding data (aleatoric) uncertainty estimated by each PI method, following the aggregation procedure described in Pearce et al. (2018) and also briefed at Section 2.2.

For **PI** estimation, we adopt two widely used text representation approaches: **TF–IDF** and **Sentence-BERT (SBERT)** (Reimers & Gurevych (2019)). TF–IDF features are extracted using 1–2 word n-grams and subsequently compressed to 128 dimensions via Truncated SVD. For **SBERT**, we employ the

`sentence-transformers/all-mpnet-base-v2` model to encode each sentence pair into embeddings $u$ and $v$, from which pairwise representations $[u, v, |u - v|, u \odot v]$ are formed. The resulting feature vectors are then $\ell_2$-normalized, projected to 256 dimensions using PCA, and cached to improve computational efficiency.

Table 7 reports the performance of the different **PI** models after careful hyperparameter tuning aimed at obtaining high-quality **PIs** with target coverage $1 - \alpha = 0.90$. To account for model uncertainty, all **PI** models are trained over 10 independent runs under identical hyperparameter settings, and data uncertainty is aggregated with the **ensemble** outputs to obtain the final uncertainty estimates. We have used the **ensemble** approach along with different **PI** estimation models namely **Quantile Regression through the Gradient Boosting (QRGB)**, **Quantile Regression Neural Network (QRNN)**, **Heteroscedastic Two-Sided (HTS) (Kendall & Gal (2017)) PI** , **Direct Uncertainty Prediction (DUP)** (Zerva et al. (2022); Zerva & Martins (2024)), **Mean Variance Estimation** (Nix & Weigend (1994)), **Mixed Density Network (MDN)** (Bishop (1994)) and proposed **Tube loss NN**. These baselines are often used for the **UQ** in **STS** task (Wang et al. (2022)). Among the evaluated approaches, the **Tube loss** based **NN** yields the best quality **PIs** for the **STS** task, outperforming the other **PI** models across both text embedding strategies.

| TF–IDF Features | | | SBERT Features | | |
|---|---|---|---|---|---|
| **Method** | **PICP** | **MPIW** | **Method** | **PICP** | **MPIW** |
| Ens–QRGB | 0.9369 | 4.4402 | Ens–QRGB | 0.9007 | 3.9463 |
| Ens–QRNN | 0.8600 | 3.3970 | Ens–HTS | 0.1864 | 0.4052 |
| Ens–HTS | 0.8738 | 3.5094 | Ens–DUP | 0.9101 | 3.4634 |
| Ens–MVE NN | 0.9594 | 5.3387 | Ens–MVE NN | 0.8811 | 2.4637 |
| Ens–MDN NN | 0.8380 | 3.3230 | Ens–MDN NN | 0.2350 | 0.4960 |
| Ens–Tube NN | **0.9289** | **4.0905** | Ens–Tube NN | **0.9152** | **2.9324** |

Table 7: Comparative analysis of the **Tube NN** against existing **PI** estimation models for the **STS** task using **TF–IDF** and **SBERT** embeddings with target coverage $1 - \alpha = 0.90$. Ens–QRGB: Ensemble Quantile Regression via Gradient Boosting; Ens–QRNN: Ensemble Quantile Regression Neural Network; Ens–HTS: Ensemble Heteroscedastic Two-Sided PI model Kendall & Gal (2017); Ens–DUP: Ensemble Direct Uncertainty Prediction Zerva et al. (2022); Zerva & Martins (2024); Ens–MVN: Ensemble Mean Variance Estimation (Nix & Weigend (1994)), Ens–MDN: Ensemble Mixture Density Network Bishop (1994).

# 5 Future Works

The **Tube loss** function evidently offers a superior method for **PI** estimation in kernel machines and NN. Although we present a natural extension of the **Tube loss** for constructing **conformal prediction sets** with finite-sample coverage guarantees, a deeper theoretical investigation of its advantages within standard and adaptive **conformal frameworks** remains an important direction for future research in view of its *recalibration* ability. Furthermore, it would also be worthwhile to implement the proposed methodology for probabilistic forecasting across a range of neural architectures employed in diverse application domains, including solar irradiance, cryptocurrency valuation, exchange rate prediction, stock market trends, ocean wave modeling, pollution estimation, and weather forecasting. In many text processing applications, the target variable is inherently multidimensional. A compelling direction for future work would be to extend the proposed prediction interval estimation methodology to accommodate such multidimensional targets.

# Acknowledgments

The authors thank Shyam Saktawat, Manush Sanchela and Harsh Sawaliya for extending their help in conducting the numerical experiments.

## Funding

This work was supported by the Smart Energy Learning Center at Dhirubhai Ambani University (Formerly DA-IICT), Gandhinagar through grant no.- CSR-25/BSES/A6-PA/SELC.

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

## A Proof of the Lemma 1

Let us assume that $\#\{y_i | y_i = \mu_2^*\} = k_1$, $\#\{y_i | y_i = r\mu_2^* + (1-r)\mu_1^*\} = k_2$, and $\#\{y_i | y_i = \mu_1^*\} = k_3$, where $\#$ represents the cardinality of a set. In other words, $k_1, k_2$ and $k_3$ represent the number of points on the boundary sets.

Thus, we obtain $n_1 + k_1 + n_2 + k_2 + n_3 + k_3 + n_4 = n$.

Let us choose, $\delta_1^* \in [0, \epsilon_1)$, $\delta_2^* \in [0, \epsilon_2)$, where $\epsilon_1$ and $\epsilon_2$ are positive real numbers, such that $\epsilon_1 < \min_{y_i < \mu_1^*}(|\mu_1^* - y_i|)$, $\epsilon_2 < \min_{y_i > \mu_2^*}|(y_i - \mu_2^*)|$, and $r\epsilon_2 + (1-r)\epsilon_1 < \min_{\mu_1^* < y_i < \mu_2^*}|y_i - (r\mu_2^* + (1-r)\mu_1^*)|$, which entails

$$\#\{y_i : \mu_2^* < y_i \le \mu_2^* + \delta_2^*\} = 0, \quad \#\{y_i : \mu_2^* < y_i \le \mu_2^* - \delta_2^*\} = 0, \quad \#\{y_i : \mu_1^* < y_i \le \mu_1^* + \delta_1^*\} = 0,$$
$$\#\{y_i : \mu_1^* < y_i \le \mu_1^* - \delta_1^*\} = 0, \quad \#\{r\mu_2^* + (1-r)\mu_1^* < y_i \le r(\mu_2^* + \delta_2^*) + (1-r)(\mu_1^* + \delta_1^*)\} = 0,$$
$$\#\{r\mu_2^* + (1-r)\mu_1^* < y_i \le r(\mu_2^* - \delta_2^*) + (1-r)(\mu_1^* - \delta_1^*)\} = 0.$$

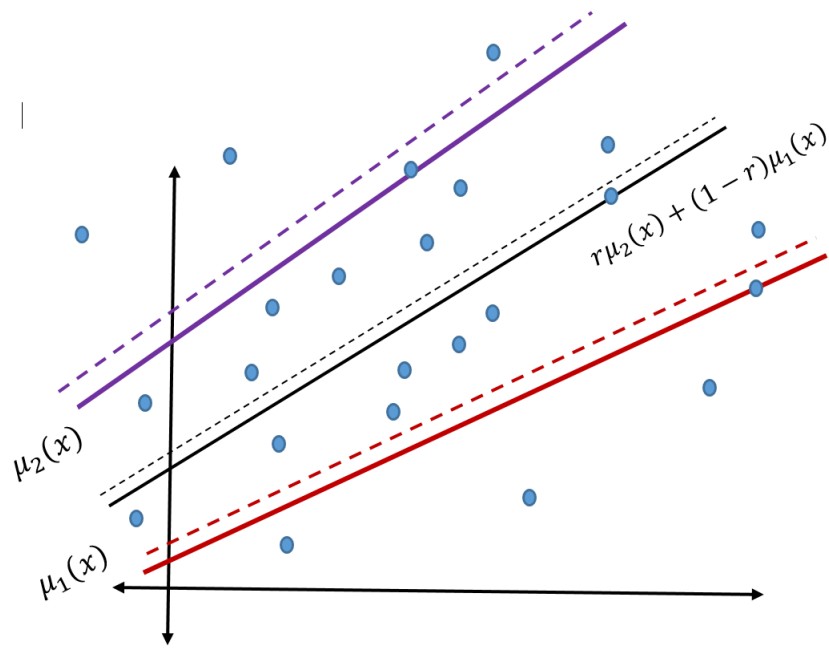

Figure 10

Given that $(\mu_1^*, \mu_2^*)$ is an optimal solution, we have $\sum_{i=1}^{n} \mathcal{L}_{1-\alpha}^r(y_i, \mu_1^* + \delta_1^*, \mu_2^* + \delta_2^*) - \sum_{i=1}^{n} \mathcal{L}_{1-\alpha}^r(y_i, \mu_1^*, \mu_2^*) \ge 0$. We now evaluate the difference of the sums for each of the following ten sets inducing a partition of $\mathbb{R}$.

$R_1 = \{y_i : y_i > \mu_2^* + \delta_2^*\}$, $R_2 = \{y_i : \mu_2^* < y_i \le \mu_2^* + \delta_2^*\}$, $R_3 = \{y_i : y_i = \mu_2^*\}$,
$R_4 = \{y_i : r(\mu_2^* + \delta_2^*) + (1-r)(\mu_1^* + \delta_1^*) < y_i < \mu_2^*\}$, $R_5 = \{y_i : r\mu_2^* + (1-r)\mu_1^* < y_i \le r(\mu_2^* + \delta_2^*) + (1-r)(\mu_1^* + \delta_1^*)\}$,
$R_6 = \{y_i : y_i = r\mu_2^* + (1-r)\mu_1^*\}$, $R_7 = \{y_i : \mu_1^* + \delta_1^* < y_i < r\mu_2^* + (1-r)\mu_1^*\}$, $R_8 = \{y_i : \mu_1^* < y_i \le \mu_1^* + \delta_1^*\}$
$R_9 = \{y_i : y_i = \mu_1^*\}$, $R_{10} = \{y_i : y_i < \mu_1^*\}$.

Denote the difference $\sum_{y_i \in R_i} \mathcal{L}_{1-\alpha}^r(y_i, \mu_1^* + \delta_1^*, \mu_2^* + \delta_2^*) - \sum_{y_i \in R_i} \mathcal{L}_{1-\alpha}^r(y_i, \mu_1^*, \mu_2^*)$ by $\Delta_i$. The simple calculation leads to the following values of $\Delta_i, i = 1, 2, ..., 10$: $\Delta_1 = -(1-\alpha)n_1\delta_2^*, \Delta_2 = 0, \Delta_3 = (\alpha)k_1\delta_2^*, \Delta_4 = (\alpha)n_2\delta_2^*, \Delta_5 = 0, \Delta_6 = k_2(\alpha)(\delta_1^* + (2-r)\mu_1^* - (1-r)\mu_2^*), \Delta_7 = -(\alpha)n_3\delta_1^*, \Delta_8 = 0, \Delta_9 = (1-\alpha)k_3\delta_1^*, \Delta_{10} = (1-\alpha)n_4\delta_1^*$.

Thus, we obtain

$$\sum_{i=1}^{n} \mathcal{L}_{1-\alpha}^{r}(y_i, \mu_1^* + \delta_1^*, \mu_2^* + \delta_2^*) - \sum_{i=1}^{n} \mathcal{L}_{1-\alpha}^{r}(y_i, \mu_1^*, \mu_2^*) = -(1-\alpha)n_1\delta_2^* + (\alpha)k_1\delta_2^* + (\alpha)n_2\delta_2^* + (\alpha)k_2(\delta_1^*$$
$$+(2-r)\mu_1^* - (1-r)\mu_2^*) - (\alpha)n_3\delta_1^* + (1-\alpha)k_3\delta_1^* + (1-\alpha)n_4\delta_1^* \geq 0. \quad (23)$$

If we assume $\delta_1^* = 0$ and $\delta_2^* > 0$, the above inequality entails

$$\frac{\alpha}{1-\alpha} \geq \frac{n_1\delta_2^*}{n_2\delta_2^* + k_1\delta_2^* + k_2((2-r)\mu_1^* - (1-r)\mu_2^*)}.$$

Given that $y_i$'s are the realizations of a continuous random variable with no discrete probability mass, $k_1/n, k_2/n \to 0$ with probability 1 as $n \to \infty$. Thus, as $n \to \infty$, with probability 1,

$$\frac{n_1}{n_2} \leq \frac{\alpha}{1-\alpha}. \quad (24)$$

.

Arguing on a similar line, if we consider $\delta_2^* = 0$ and $\delta_1^* > 0$ in (23), then we have the following inequality

$$\frac{(k_3 + n_4)\delta_1^*}{n_3\delta_1^* - k_2(\delta_1^* + (2-r)\mu_1^* - (1-r)\mu_2^*)} \geq \frac{\alpha}{1-\alpha}$$

As $n \to \infty$, we can state that the following inequality holds with probability 1,

$$\frac{n_4}{n_3} \geq \frac{\alpha}{1-\alpha}. \quad (25)$$

Again, starting with $(\mu_1^*, \mu_2^*)$ as an optimal solution, we have $\sum_{i=1}^{m} \mathcal{L}_{1-\alpha}^{r}(y_i, \mu_1^* - \delta_1^*, \mu_2^* - \delta_2^*) - \sum_{i=1}^{m} \mathcal{L}_{1-\alpha}^{r}(y_i, \mu_1^*, \mu_2^*) \geq 0$. For computing this, let us evaluate the difference of the sums for each of the following ten sets inducing a partition of $\mathbb{R}$.

$R_1' = \{y_i : y_i > \mu_2^*\}$, $R_2' = \{y_i : y_i = \mu_2^*\}$, $R_3' = \{y_i : \mu_2^* - \delta_2^* \leq y_i < \mu_2^*\}$, $R_4' = \{y_i : r\mu_2^* + (1-r)\mu_1^* < y_i < \mu_2^* - \delta_2^*\}$, $R_5' = \{y_i : y_i = r(\mu_2^*) + (1-r)\mu_1^*\}$, $R_6' = \{y_i : r(\mu_2^* - \delta_2^*) + (1-r)(\mu_1^* - \delta_1^*) \leq y_i < r\mu_2^* + (1-r)\mu_1^*\}$, $R_7' = \{\mu_1^* < y_i < r(\mu_2^* - \delta_2^*) + (1-r)(\mu_1^* - \delta_1^*)\}$, $R_8' = \{y_i : y_i = \mu_1^*\}$, $R_9' = \{y_i : \mu_1^* < y_i \leq \mu_1^* - \delta_1^*\}$, $R_{10}' = \{y_i : y_i < \mu_1^* - \delta_1^*\}$.

Denoting $\sum_{y_i \in R_i} \mathcal{L}_{1-\alpha}^{r}(y_i, \mu_1^* - \delta_1^*, \mu_2^* - \delta_2^*) - \sum_{y_i \in R_i} \mathcal{L}_{1-\alpha}^{r}(y_i, \mu_1^*, \mu_2^*)$ by $\Delta_i'$ we obtain the following values of $\Delta_i', i = 1, 2, ..., 10$: $\Delta_1' = (1-\alpha)n_1\delta_2^*$, $\Delta_2' = -(\alpha)k_1\delta_2^*$, $\Delta_3' = 0$, $\Delta_4' = -(\alpha)n_2\delta_2^*$, $\Delta_5' = -(\alpha)k_2\delta_2^*$, $\Delta_6' = 0$, $\Delta_7' = (\alpha)n_3\delta_1^*$, $\Delta_8' = (\alpha)k_3\delta_1^*$, $\Delta_9' = 0$, $\Delta_{10}' = -(1-\alpha)n_4\delta_1^*$.

Thus, we obtain

$$\sum_{i=1}^{n} \mathcal{L}_{1-\alpha}^{r}(y_i, \mu_1^* + \delta_1^*, \mu_2^* + \delta_2^*) - \sum_{i=1}^{n} \mathcal{L}_{1-\alpha}^{r}(y_i, \mu_1^*, \mu_2^*) = (1-\alpha)n_1\delta_2^* - (\alpha)k_1\delta_2^* - (\alpha)n_2\delta_2^* - (\alpha)k_2\delta_2^*$$
$$+(\alpha)n_3\delta_1^* + (\alpha)k_3\delta_1^* - (1-\alpha)n_4\delta_1^* \geq 0. \quad (26)$$

Now, if we assume $\delta_1^* = 0$ and $\delta_2^* > 0$ in (26), then we obtain,

$$\frac{n_1}{k_1 + n_2 + k_2} \geq \frac{\alpha}{1-\alpha},$$

which entails

$$\frac{n_1}{n_2} \geq \frac{\alpha}{1-\alpha}, \quad (27)$$

as $n \to \infty$ with probability 1.

Similarly, considering $\delta_2^* = 0$ and $\delta_1^* > 0$ in (26), we obtain as $n \to \infty$,

$$\frac{n_4}{n_3} \leq \frac{\alpha}{1-\alpha}, \tag{28}$$

holds with probability 1.

Thus, (24) and (27) imply that as $n \to \infty$

$$\frac{n_1}{n_2} = \frac{\alpha}{1-\alpha} \tag{29}$$

holds with probability 1.

Similarly, combining (25) and (28), we obtain as $n \to \infty$,

$$\frac{n_4}{n_3} = \frac{\alpha}{1-\alpha} \tag{30}$$

holds with probability 1.

Thus, combining (29) and (30), we obtain, as $n \to \infty$,

$$\frac{n_1 + n_4}{n_2 + n_3} = \frac{\alpha}{1-\alpha} \tag{31}$$

holds with probability 1.

## B Gradient Descent method for Tube loss-based kernel machines

The Tube loss-based kernel machine estimates a pair of functions

$$\mu_1(x) := \sum_{i=1}^{n} k(x_i, x)\eta_i + \eta_0 \quad \text{and} \quad \mu_2(x) := \sum_{i=1}^{n} k(x_i, x)\beta_i + \beta_0. \tag{32}$$

where $k(x, y)$ is positive definite kernel. For the sake of simplicity, we rewrite $\mu_1(x)$ and $\mu_2(x)$ in vector form

$$\mu_1(x) := K(A^T, x)\eta + \eta_0 \quad \text{and} \quad \mu_2(x) := K(A^T, x)\beta + \beta_0. \tag{33}$$

where $A$ is the $n \times p$ data matrix containing $n$ training points in $\mathbb{R}^p$, $K(A^T, x) = \left[k(x_1, x), k(x_2, x), .., k(x_n, x)\right]$,

$\eta = \begin{bmatrix} \eta_1 \\ \eta_2 \\ .. \\ \eta_n \end{bmatrix}$ and $\beta \begin{bmatrix} \beta_1 \\ \beta_2 \\ .. \\ \beta_n \end{bmatrix}$. The Tube loss-based kernel machines considers the problem

$$\min_{(\eta, \beta, \eta_0, \beta_0)} J_2(\eta, \beta, \eta_0, \beta_0) = \frac{\lambda}{2}(\eta^T \eta + \beta^T \beta) + \sum_{i=1}^{n} \mathcal{L}_{1-\alpha}^r \left(y_i, \left(K(A^T, x_i)\eta + \eta_0\right), \left(K(A^T, x_i)\beta + \beta_0\right)\right)$$

$$+ \delta \sum_{i=1}^{n} \left|\left(K(A^T, x_i)(\beta - \eta) + (\beta_0 - \eta_0)\right)\right|, \tag{34}$$

where $\mathcal{L}_{1-\alpha}^r$ is the Tube loss function as given in (12) with the parameter $r$.

For a given point $(x_i, y_i)$, let us compute the gradient of $\mathcal{L}^r_{1-\alpha}\big(y_i, \big(K(A^T, x_i)\eta + b_1\big), \big(K(A^T, x_i)\beta + b_2\big)\big)$ first.

$$\frac{\partial \mathcal{L}^r_{1-\alpha}\left(y_i, \left(K(A^T, x_i)\eta + \eta_0\right), \left(K(A^T, x_i)\beta + \beta_0\right)\right)}{\partial \beta} =$$

$$\begin{cases} \alpha K(A, x_i), & \text{if } (K(A^T, x_i)\eta + \eta_0) < y_i < (K(A^T, x_i)\beta + \beta_0) \text{ and } y_i > (K(A^T, x_i)(r\beta + (1-r)\eta) + (r\beta_0 + (1-r)\eta_0). \\ 0, & \text{if } (K(A^T, x_i)\eta + \eta_0) < y_i < (K(A^T, x_i)\beta + \beta_0) \text{ and } y_i < (K(A^T, x_i)(r\beta + (1-r)\eta) + (r\beta_0 + (1-r)\eta_0). \\ 0, & \text{if } K(A^T, x_i)\eta + \eta_0 > y. \\ -(1-\alpha)K(A, x_i), & \text{if } K(A^T, x_i)\beta + \beta_0 < y. \end{cases}$$

$$\frac{\partial \mathcal{L}^r_{1-\alpha}\left(y_i, \left(K(A^T, x_i)\eta + \eta_0\right), \left(K(A^T, x_i)\beta + \beta_0\right)\right)}{\partial \beta_0} =$$

$$\begin{cases} \alpha, & \text{if } (K(A^T, x_i)\eta + \eta_0) < y_i < (K(A^T, x_i)\beta + \beta_0) \text{ and } y_i > (K(A^T, x_i)(r\beta + (1-r)\eta) + (r\beta_0 + (1-r)\eta_0). \\ 0, & \text{if } (K(A^T, x_i)\eta + \eta_0) < y_i < (K(A^T, x_i)\beta + \beta_0) \text{ and } y_i < (K(A^T, x_i)(r\beta + (1-r)\eta) + (r\beta_0 + (1-r)\eta_0). \\ 0, & \text{if } K(A^T, x_i)\eta + \eta_0 > y. \\ -(1-\alpha), & \text{if } K(A^T, x_i)\beta + \beta_0 < y. \end{cases}$$

$$\frac{\partial \mathcal{L}^r_{1-\alpha}\left(y_i, \left(K(A^T, x_i)\eta + \eta_0\right), \left(K(A^T, x_i)\beta + \beta_0\right)\right)}{\partial \eta} =$$

$$\begin{cases} 0, & \text{if } (K(A^T, x_i)\eta + \eta_0) < y_i < (K(A^T, x_i)\beta + \beta_0) \text{ and } y_i > (K(A^T, x_i)(r\beta + (1-r)\eta) + (r\beta_0 + (1-r)\eta_0). \\ -\alpha K(A, x_i), & \text{if } (K(A^T, x_i)\eta + \eta_0) < y_i < (K(A^T, x_i)\beta + \beta_0) \text{ and } y_i < (K(A^T, x_i)(r\beta + (1-r)\eta) + (r\beta_0 + (1-r)\eta_0). \\ (1-\alpha)K(A, x_i), & \text{if } K(A^T, x_i)\eta + \eta_0 > y. \\ 0 & \text{if } K(A^T, x_i)\beta + \beta_0 < y. \end{cases}$$

$$\frac{\partial \mathcal{L}^r_{1-\alpha}\left(y_i, \left(K(A^T, x_i)\eta + \eta_0\right), \left(K(A^T, x_i)\beta + \beta_0\right)\right)}{\partial \eta_0} =$$

$$\begin{cases} 0, & \text{if } (K(A^T, x_i)\eta + \eta_0) < y_i < (K(A^T, x_i)\beta + \beta_0) \text{ and } y_i > (K(A^T, x_i)(r\beta + (1-r)\eta) + (r\beta_0 + (1-r)\eta_0). \\ -\alpha, & \text{if } (K(A^T, x_i)\eta + \eta_0) < y_i < (K(A^T, x_i)\beta + \beta_0) \text{ and } y_i < (K(A^T, x_i)(r\beta + (1-r)\eta) + (r\beta_0 + (1-r)\eta_0). \\ (1-\alpha), & \text{if } K(A^T, x_i)\eta + \eta_0 > y. \\ 0 \quad , & \text{if } K(A^T, x_i)\beta + \beta_0 < y. \end{cases}$$

For a data point $(x_k, y_k)$ that lies exactly on the boundary of the PI or upon the surface $r\big(K(A^T, x_k)\beta + \beta_0\big) + (1-r)\big(K(A^T, x_k)\eta + \eta_0\big)$, the Tube loss is not differentiable. In this case, the gradient is not unique, and any valid sub-gradient may be used for computation.

Now, for given data point $(x_i, y_i)$, we consider the width of PI tube $\delta\big|(K(A^T, x_i)(\beta - \eta) + (\beta_0 - \eta_0)\big|$ and denote it as $J(\beta, \eta, \beta_0, \eta_0, x_i)$ and compute

$$\frac{\partial J(\beta, \eta, \beta_0, \eta_0, x_i)}{\partial \beta} = sign((K(A^T, x_i)(\beta - \eta) + (\beta_0 - \eta_0))K(A, x_i) \quad \text{if } (K(A^T, x_i)(\beta - \eta) + (\beta_0 - \eta_0) \neq 0$$

$$\frac{\partial J(\beta, \eta, \beta_0, \eta_0, x_i)}{\partial \beta_0} = sign((K(A^T, x_i)(\beta - \eta) + (\beta_0 - \eta_0)) \quad \text{if } (K(A^T, x_i)(\beta - \eta) + (\beta_0 - \eta_0) \neq 0$$

$$\frac{\partial J(\beta, \eta, \beta_0, \eta_0, x_i)}{\partial \eta} = -sign((K(A^T, x_i)(\beta - \eta) + (\beta_0 - \eta_0))K(A, x_i) \quad \text{if } (K(A^T, x_i)(\beta - \eta) + (\beta_0 - \eta_0) \neq 0$$

$$\frac{\partial J(\beta, \eta, \beta_0, \eta_0, x_i)}{\partial \eta_0} = -sign((K(A^T, x_i)(\beta - \eta) + (\beta_0 - \eta_0)) \quad \text{if } (K(A^T, x_i)(\beta - \eta) + (\beta_0 - \eta_0) \neq 0$$

For a data point $(x_i, y_i)$ satisfying $K(A^T, x_i)(\beta - \eta) + (\beta_0 - \eta_0) = 0$ the function $J(\beta, \eta, \beta_0, \eta_0; x_i)$ is not differentiable. In this case, the gradient is not unique, and any valid sub-gradient may be used for computation..

Now, we state gradient descent algorithm for the Tube loss based kernel machine problem.
**Algorithm 2:-**

Input:- Training Set $T = \{(x_i, y_i) : x_i \in \mathbb{R}^p, y_i \in \mathbb{R}, i = 1, 2, ...n\}$, target coverage $1 - \alpha \in (0, 1)$, shift parameter $r$, re-calibration parameter $\delta$, learning rate $\gamma$ and *tol*.
Initialize:- $\beta^0, \eta^0 \in \mathbb{R}^n$ and $\beta_0^0, \eta_0^0 \in \mathbb{R}$.

Repeat

$$\beta^{(k+1)} = \beta^{(k)} - \gamma_k \Big(\lambda \beta^{(k)} \sum_{i=1}^{n} \frac{\partial \mathcal{L}_{1-\alpha}^r \Big(y_i, \big(K(A^T, x_i)\beta + \beta_0\big), \big(K(A^T, x_i)\eta + \eta_0\big)\Big)}{\partial \beta^{(k)}} + \delta \sum_{i=1}^{n} \frac{\partial J(\beta, \eta, \beta_0, \eta_0, x_i)}{\partial \beta^{(k)}} \Big)$$

$$\beta_0^{(k+1)} = \beta_0^{(k)} - \gamma_k \Big( \sum_{i=1}^{n} \frac{\partial \mathcal{L}_{1-\alpha}^r \Big(y_i, \big(K(A^T, x_i)\beta + \beta_0\big), \big(K(A^T, x_i)\eta + \eta_0\big)\Big)}{\partial \beta_0^{(k)}} + \delta \sum_{i=1}^{n} \frac{\partial J(\beta, \eta, \beta_0, \eta_0, x_i)}{\partial \beta_0^{(k)}} \Big)$$

$$\eta^{(k+1)} = \eta^{(k)} - \gamma_k \Big(\lambda \eta^{(k)} \sum_{i=1}^{n} \frac{\partial \mathcal{L}_{1-\alpha}^r \Big(y_i, \big(K(A^T, x_i)\beta + \beta_0\big), \big(K(A^T, x_i)\eta + \eta_0\big)\Big)}{\partial \eta^{(k)}} + \delta \sum_{i=1}^{n} \frac{\partial J(\beta, \eta, \beta_0, \eta_0, x_i)}{\partial \eta^{(k)}} \Big)$$

$$\eta_0^{(k+1)} = \eta_0^{(k)} - \gamma_k \Big( \sum_{i=1}^{n} \frac{\partial \mathcal{L}_{1-\alpha}^r \Big(y_i, \big(K(A^T, x_i)\beta + \beta_0\big), \big(K(A^T, x_i)\eta + \eta_0\big)\Big)}{\partial \eta_0^{(k)}} + \delta \sum_{i=1}^{n} \frac{\partial J(\beta, \eta, \beta_0, \eta_0, x_i)}{\partial \eta_0^{(k)}} \Big)$$

Until $\left\| \begin{bmatrix} \beta^{(k+1)} - \beta^{(k)} \\ \beta_0^{(k+1)} - \beta_0^{(k)} \\ \eta^{(k+1)} - \eta^{(k)} \\ \eta_0^{(k+1)} - \eta_0^{(k)} \end{bmatrix} \right\|_2 \geq tol.$

In our implementation [1], the gradient descent algorithm for the Tube loss kernel machine initially utilizes only the gradient of the Tube loss function. After a certain number of iterations, once we confirm that the upper bound of the PI, $K(A^T x_i)\beta + \beta_0$, has moved above the lower bound, $K(A^T x_i)\eta + \eta_0$, of PI, we incorporate the gradient of the PI tube width into the training of the Tube loss kernel machine.

## C   Additional Experimental Details

**Tuning $r$ parameter:-** Ideally, the tuning of the $r$ parameter in the tube loss function should align with the skewness of the distribution $y|x$. However, real-world datasets often exist in high dimensions. Consequently, for a specific value of $x$, there may be only a few of $y_i$ values in the training dataset, leading to potential inaccuracies in estimating the skewness of the distribution $y|x$. In practical benchmark experiments, we have tuned the $r$ values for the tube loss machine from the set $\{0.1, 0.2, ..., 0.9\}$.

**Tuning $\delta$ parameter:-** Initially, we initialize $\delta$ to zero in the Tube loss-based machine for benchmark dataset experiments and aim to achieve the narrowest PI by adjusting the $r$ parameter, either moving the PI tube upwards or downwards. Once the $r$ parameter is set and if the Tube loss-based machine achieves a coverage higher than the target $t$ on the validation set, we gradually fine-tune the $\delta$ parameter from the set $\{0.001, 0.005, 0.1, 0.15, 0.2\}$ to minimize the tube width while maintaining the desired target coverage $t$.

For the numerical results in Table 2, the tuned parameters for the Tube Loss are summarized in Table 8.

| dataset | r (tube_r) | delta (penalty) | lr | batch | dropout | epochs |
|---------|-----------|-----------------|------|-------|---------|--------|
| boston | 0.5 | 0.03 | 0.05 | 10000 | 0.2 | 400 |
| concrete | 0.25 | 0.05 | 0.015 | 64 | 0.25 | 150 |
| cpu_act | 0.5 | 0.03 | 0.05 | 10000 | 0.2 | 400 |
| energy | 0.9 | 0.01 | 0.05 | 10000 | 0.2 | 400 |
| kin8nm | 0.1 | 0.01 | 0.05 | 10000 | 0.2 | 400 |
| miami | 0.3 | 0.05 | 0.05 | 10000 | 0.2 | 400 |
| naval | 0.3 | 0.05 | 0.05 | 10000 | 0.2 | 400 |
| power | 0.3 | 0.1 | 0.05 | 10000 | 0.2 | 400 |
| protein | 0.1 | 0.01 | 0.05 | 10000 | 0.2 | 400 |
| sulfur | 0.1 | 0.01 | 0.05 | 10000 | 0.2 | 400 |
| wine | 0.5 | 0.05 | 0.05 | 10000 | 0.2 | 400 |
| yacht | 0.5 | 0.03 | 0.01 | 10000 | 0.1 | 400 |

Table 8: Best hyper-parameters for Tube Loss based NN for results generated in Table 2.

---

[1]A MATLAB implementation of gradient descent with Tube loss is provided in supplementary material code.

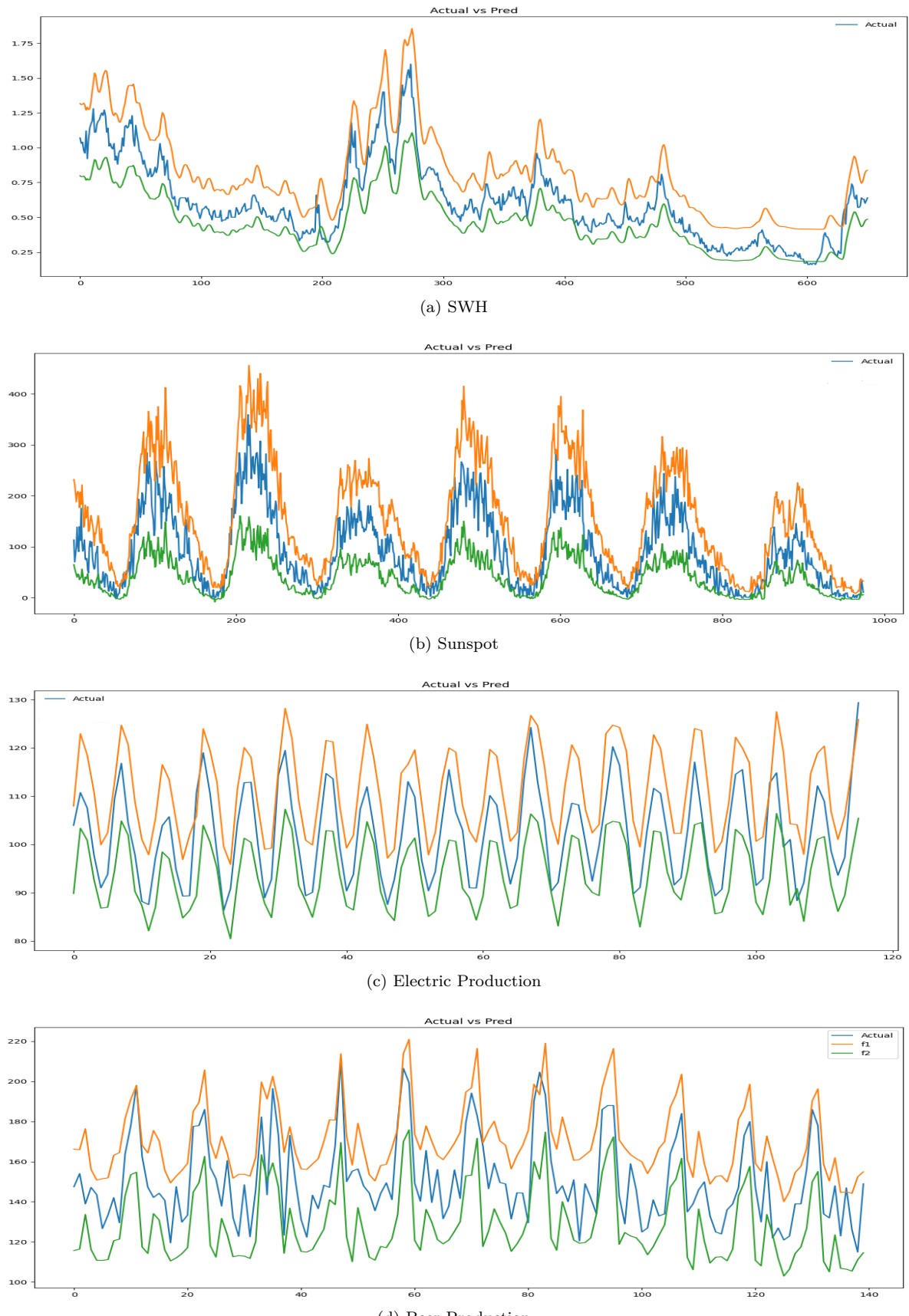

Figure 11: Probabilistic forecast of LSTM with proposed Tube loss function.

| Dataset (Size) | LSTM Structure | Batch Size | | Up,Low | r,δ | Window Size | Learning Rate | |
|---|---|---|---|---|---|---|---|---|
| | | Q- LSTM | T-LSTM | Q- LSTM | T-LSTM | | Q-LSTM | T-LSTM |
| Sunspots (3265) | [256(0.3) ,128(0.2)] | 128 | 128 | 0.98,0.03 | 0.5,0 | 16 | 0.001 | 0.001 |
| Electric Production (397) | [64] | 32 | 16 | 0.98,0.03 | 0.5,0.01 | 12 | 0.01 | 0.001 |
| Daily Female Birth (365) | [100] | 64 | 128 | 0.98,0.03 | 0.5,0.01 | 12 | 0.01 | 0.005 |
| SWH (2170) | [128(0.4),64(0.3),32(0.2)] | 300 | 300 | 0.98, 0.03 | 0.5, 0.01 | 100 | 0.001 | 0.001 |
| Temperature (3651) | [16,8] | 300 | 300 | 0.98, 0.03 | 0.5, 0.01 | 100 | 0.001 | 0.0001 |
| Beer Production (464) | [64,32] | 64 | 64 | 0.98, 0.03 | 0.1, 0.01 | 12 | 0.001 | 0.0001 |

Table 9: Tuned parameter values for Q-LSTM and T-LSTM model for considered benchmark datasets in Table 3. The LSTM architecture [128(0.4),64(0.3),32(0.2)] means that three hidden layers with neurons 128, 64 and 32 respectively and each hidden layer is followed by a drop out layer which are 0.4,0.3 and 0.2 respectively.

