# Response to Reviewer (Tube Loss: A Novel Loss Function for Prediction Interval Estimation)

The authors gratefully acknowledge the reviewers for their thoughtful and constructive comments. We believe that careful consideration of these suggestions has meaningfully strengthened the quality and clarity of our work. We now present our point-by-point responses to the reviewers' observations and recommendations. A copy of the revised manuscript is attached with this response, where all changes have been highlighted in blue for ease of reference.

## 1 Reviewer RuFc

Comment 1:- Regarding the computational time of the quantile and Tube loss-based model:-

(i) *The novelty appears incremental. The motivation is to get both bounds from one optimization, but in practice training two separate quantile models is already standard and computationally efficient. In a related spirit, the proposed method needs a grid search for the hyperparameter r through multiple runs, so it is also computationally expensive and not a single optimization as claimed.*

(ii) *The claim of "single optimization" is misleading. The proposed method requires a grid search for the hyperparameter through runs, which makes the total training cost higher than standard quantile regression. Computation-time reporting is not comparable as presented.*

(iii) *For a fair comparison, the paper should report end-to-end computation time including hyperparameter tuning, especially since the proposed method appears to require extensive tuning across multiple runs.*

**Response:-** We sincerely apologize for the confusion probably created by our statement on page 3, lines 21-23. To clear the confusion in the revised manuscript, we have slightly changed the statement as follows:

"A few questions arise naturally at this point: Given q, can the lower and upper bounds be estimated simultaneously by solving a single optimization problem? In other words, for a given value of q, is it possible to obtain the PI or both the quantiles as a direct output of a single optimization process?"

What we intended to convey is that the Tube-loss–based framework solves a single optimization problem to jointly learn the lower and upper bounds of the prediction interval, whereas quantile-based PI approaches typically require solving two separate optimization problems to estimate the respective bounds. Following this line of argument, several loss functions have recently been proposed, viz., the LUBE loss Khosravi et al. (2011), the QD loss Pearce et al. (2018), its improved variant QD$^+$ Salem et al. (2020), and the RQR loss Pouplin et al. (2024).

| Method | Shift Parameter | MPIW | PICP | #Parameters | Time(sec) |
|---|---|---|---|---|---|
| Tube Loss NN | $r = 0.5$ | 10.536 | 0.796 | 800 | 4.18 |
| Tube Loss NN | $r = 0.625$ | 12.839 | 0.806 | 800 | 4.30 |
| Tube Loss NN | $r = 0.375$ | 7.294 | 0.814 | 800 | 4.35 |
| Tube Loss NN | $r = 0.25$ | **6.851** | **0.800** | 800 | 4.27 |
| Tube Loss NN | $r = 0.125$ | 7.073 | 0.806 | 800 | 4.94 |
| QRNN | $q = 0.100, 1 + q - \alpha = 0.900$ | 7.887 | 0.812 | 1200 | 7.60 |
| QRNN | $q = 0.125, 1 + q - \alpha = 0.9125$ | 9.027 | 0.800 | 1200 | 7.47 |
| QRNN | $q = 0.075, 1 + q - \alpha = 0.875$ | 7.731 | 0.810 | 1200 | 7.59 |
| QRNN | $q = 0.050, 1 + q - \alpha = 0.850$ | 7.071 | 0.794 | 1200 | 7.54 |
| QRNN | $q = 0.025, 1 + q - \alpha = 0.825$ | **7.076** | **0.800** | 1200 | 7.53 |

Table 1: Computation time of Tube NN and QRNN based PI with confidence level $1 - \alpha = 0.80$.

As suggested by the reviewer, in the following, we report the end-to-end computation times of the two methods, viz., Quantile regression based and Tube loss–based NN models for **PI** estimation, including hyperparameter tuning, by running experiments on a toy data set. We appreciate the reviewer's suggestion. If the reviewer believes that including this discussion would strengthen the manuscript, we would be happy to incorporate it in the revised version.

We generate a synthetic regression dataset $\{(x_i, y_i)\}_{i=1}^{500}$. The inputs $x_i$ are independently sampled from a uniform distribution over $[-2\pi, 2\pi]$, and the outputs are produced according to

$$y_i = \frac{\sin x_i}{x_i} + \epsilon_i, \tag{1}$$

where $\epsilon_i = \epsilon_i^* - 5$, with $\epsilon_i^*$ independently drawn from a $\chi^2$ distribution with 5 degrees of freedom. A validation set of size 500 is created using the same data-generating process. We investigate **PI** estimation in a neural network setting with target coverage $1 - \alpha = 0.8$ using two approaches: (i) Tube Loss based, referred to as TUBE NN, and (ii) quantile regression based, referred to as QRNN. To obtain high-quality **PI**s, the hyperparameter $r$ must be tuned for TUBE NN, while the quantile level $q$ must be specified for QRNN.

For conducting the experiments, we first fix the NN hyperparameters by training a mean regression model. Specifically, we adopt a fixed network architecture with a single hidden layer of 200 neurons, a batch size of 40, the Adam optimizer, a learning rate of 0.02, and 200 training epochs. Using this fixed architecture, we train both the QRNN and Tube NN models to construct PIs.

In the QRNN framework, the shifting of the PI is controlled by the quantile level $q$, which must be tuned to align the PI with the densest region of the data in order to obtain smaller MPIW values. Similarly, in the Tube NN framework, the parameter $r$ governs the relative positioning of the PI and must be tuned accordingly. For a fair comparison, we evaluate the PI quality and the tuning cost of both methods over five different values of $r$ and $q$. The numerical result is listed in Table 1 in this response letter.

For tuning the hyperparameter $r$, we first train the TUBE NN with the default setting $r = 0.5$ and assess the quality of resulting **PI**s on the test set using MPIW and PICP. We then repeat this procedure for $r \in \{0.625, 0.375, 0.25, 0.125\}$, ultimately selecting $r = 0.125$ as the best performing

setting. On the other hand to tune the quantile parameter $q$ in QRNN, we start with the default value $q = 0.10$, which yields a centered **PI**, and evaluate MPIW and PICP on the test set. This procedure is subsequently repeated for $q \in \{0.125, 0.075, 0.050, 0.025\}$, leading to the selection of $q = 0.025$ as the best performing setting.

It should be noted that the overall hyperparameter tuning time for QRNN amounts to 37.73 seconds, while TUBE NN requires only 22.04 seconds, indicating that QRNN incurs approximately **71%** higher tuning cost. Furthermore, decreasing the value of $r$ in TUBE NN and the quantile level $q$ in QRNN leads to improved **PI** quality, which is consistent with expectations given the positively skewed nature of the conditional distribution $y \mid x$.

Comment 2:- Regarding the comparison with the conformal regression:-

(i) *Conformal prediction is a state-of-the-art baseline in uncertainty quantification, while the authors excludes them due to reasons like different framework or computation cost.*

(ii) *Compare with the state-of-the-art conformal baseline.*

(iii) *The authors claimed that conformal prediction requires retraining from scratch. However, standard methods are efficient post-hoc steps that do not require retraining.*

**Response:-** We sincerely thank the reviewer for the insightful comments and constructive suggestions. We concur that conformal regression constitutes a widely adopted and powerful framework for uncertainty quantification in regression settings. In response to the reviewer's recommendation, we have incorporated an additional empirical study in Section 4.5, wherein we compare the performance of the Tube Loss–based conformal prediction method with the quantile regression–based conformal approach proposed by Romano et al. (2019) across nine widely used benchmark datasets.

The empirical results indicate that the Tube Loss–based approach yields high-quality conformal prediction sets with competitive performance. That said, a rigorous theoretical explanation supporting these empirical observations remains an open problem. We acknowledge this limitation and intend to pursue a detailed theoretical investigation in future work.

Comment 3:- Provide the stronger theoretical or empirical justification for the proposed method.

**Response:-** To further reinforce the empirical validation of uncertainty quantification using the Tube Loss–based **PI**, we have augmented the revised manuscript with two additional subsections in the **Experiments** section (Section 4). Specifically, Subsection 4.5 investigates the integration of Tube Loss within the conformal prediction framework, while Subsection 4.6 demonstrates its applicability to the Semantic Textual Similarity (STS) task, a regression problem that is fundamentally distinct from the regression settings examined earlier.

We have already provided a brief account of the empirical evaluation of Tube Loss–based conformal prediction in our response to Comment 2. For the STS application, we employ two widely used text representation techniques, namely TF–IDF and Sentence–BERT (SBERT) Reimers & Gurevych (2019). To capture *model uncertainty*, we adopt the deep ensemble strategy Lakshminarayanan et al. (2017), whereas *data uncertainty* is addressed through several prediction interval (PI) estimation methods, including Quantile Regression via Gradient Boosting (QRGB), Quantile Regression Neural Networks (QRNN), Heteroscedastic Two-Sided (HTS) PIs Kendall & Gal (2017), Direct Uncertainty Prediction (DUP) Zerva et al. (2022); Wang et al. (2022), Mean Variance Estimation (MVE) NN Nix

& Weigend (1994), Mixture Density Networks (MDN) Bishop (1994), and the proposed Tube Loss–based neural network with ensembles.

Across both embedding schemes and under a common experimental setup, the Tube Loss–based ensemble NN consistently produces sharper PIs while preserving the nominal coverage level. These findings indicate that the proposed framework is not confined to time-series settings but extends naturally to structurally distinct regression tasks involving high-dimensional and unstructured data.

The revised manuscript contains the overall comparative analysis of the Tube loss PI model with 10 different PI estimation method available in the literature in the distribution- free and distribution based setting. For probabilistic forecast task, the Tube loss based neural architectures has been compared with the 9 recent probabilistic forecasting neural models.

## 2 Reviewer QADY

- The paper has some merit. However, in my own opinion, it requires significant changes.

- There are many grammatical errors. For example, on page 6, there is one sentence, "... some good theoretical properties, such as, it attains coverage, and it is unbiased and consistent", containing multiple grammatical issues. It is the authors' responsibility to make a significant effort to improve the writing.

- There are also many notation inconsistencies. For example, again on page 6, equation (12), there is no input for. I also highly recommend that the authors make some effort to make the mathematical notation used in the paper aesthetically appealing. For instance, $\mathcal{L}_{LUBE}$. looks quite annoying.

**Response:-** We thank the reviewer for the appreciation of our work and sincerely apologize for the inconvenience caused by the grammatical and spelling errors, as well as the notation inconsistencies in the original submission. In response, we have carefully revised the manuscript to correct all grammatical and typographical issues and to ensure uniform notation throughout. The entire manuscript has been thoroughly proofread to eliminate any remaining inconsistencies. To the best of our knowledge, the revised version is now free of spelling and grammatical errors, and the notation has been standardized and made consistent across all sections. For ease of reference, all modifications in the manuscript are highlighted in blue.

All loss functions are now explicitly stated together with their respective arguments. Ambiguous or missing inputs (including the issue in Equation (12)) have been corrected. Furthermore, the notation has been standardized throughout the manuscript to avoid reuse of symbols with multiple meanings. We have also simplified and refined several expressions to enhance readability and visual clarity.

**Comment 2:-** In Figure 1, what is $t$ ? A visual comparison between the tube loss and the RQR loss is more recommended than displaying the RQR loss in almost the end of the paper.

**Reply:-**

These issues have been addressed in the revised manuscript. In particular, the target coverage is now consistently denoted by $1 - \alpha$ throughout. We have moved the plot of the RQR loss to the Related Work section (see Figure 1). Moreover, we have reformulated the RQR loss expression (cf. Eq. (9)) to align with the Tube loss formulation, thereby making their structural similarities and differences more explicit. As a result, readers can now directly compare the RQR loss illustrated in

Figure 1 with the Tube loss presented in Figure 2 in the revised manuscript.

**Comment 3:-** The theoretical results should be stated with more care. For instance, in both Lemma 1 and Proposition 1, some of the notations are not defined in the paper, such as $n_1, n_2, n_3, ..n_4$.

**Reply:-** We thank the reviewer for the suggestion and have carefully standardized the symbols and notation while presenting the theoretical results. In Lemma 1 and Proposition 1, for mathematical convenience and clarity in subsequent discussions, we explicitly state that $n_k$ denotes the number of data points in $\Re_k(\mu_1^*, \mu_2^*)$, for $k = 1, 2, 3, 4$ ahead of stating the Lemma 1.

**Comment 4:-** In section 3.3, what is $\mathcal{M}$ ? Should it be $\mathcal{L}$ instead?

**Response:-** Yes, and we acknowledge this error on our part. These issues have now been addressed, and the corresponding corrections have been incorporated in the revised manuscript.

**Comment 5:-** Numerical comparisons should be more comprehensive. In the current version, the tube loss-based interval is only compared against one or two alternative methods.

**Response:-** We thank the reviewer for the suggestion. In the revised manuscript, we provide a comprehensive comparison of the Tube loss–based PI model with 10 existing PI estimation methods spanning both distribution-free and distribution-based frameworks. For probabilistic forecasting tasks, the proposed Tube loss neural architectures are additionally compared with 9 recent probabilistic forecasting models.

While the earlier version included comparisons primarily with distribution-free methods viz., RQR loss NN (Pouplin et al. (2024)), QRNN, SQR-C and SQR-N (Tagasovska & Lopez-Paz (2019)), and QD loss NN (Pearce et al. (2018)), the revised manuscript extends the evaluation to distribution-based approaches, including Mean Variance Estimation (MVE) NN (Nix & Weigend (1994)), Heteroscedastic Two-Sided (HTS) (Kendall & Gal (2017)), and Mixture Density Networks (MDN) (Bishop (1994)), Quantile Regression via Gradient Boosting (QRGB) and Direct Uncertainty Prediction (DUP) (Zerva et al. (2022); Wang et al. (2022)) on STS task estimation with two different text embedding, S-BERT and TF-IDF in Table 7.

Moreover, Section 4 has been expanded with two new subsections: Subsection 4.5 studies the integration of Tube Loss within the conformal prediction framework and compares it with its quantile-based counterpart, while Subsection 4.6 demonstrates its applicability to the STS task, a regression setting structurally different from the tabular benchmarks considered earlier.

**Comment 6:-** The theoretical results are somewhat preliminary. This is not a very serious issue for TMLR. But I would like to see that the authors could discuss more caveats about the theoretical results. For instance, when gradient-based methods are used, even though the conditional density satisfies certain smoothness conditions, it is not necessarily the case that the eventual output converges to the true quantile regression functions, and thus the cited results of Takeuchi et al. may not be directly applicable. Proposition 1 should therefore be stated with more caution.

**Response:-**

We thank the reviewer for this valuable suggestion. Proposition 1 has been revised to explicitly impose regularity conditions on the function class $\mathcal{F}$ from which $\mu_1(x)$ and $\mu_2(x)$ are selected, requiring $\mathcal{F}$ to be measurable and sufficiently regular to ensure uniform convergence of empirical to population risk (See page 10 of the revised manuscript). We also clarify that, analogous to

pinball loss minimization, the Tube loss minimizer guarantees only *asymptotic average calibration*, $P(\mu_1(x) \le y \le \mu_2(x)) \to 1 - \alpha$, which does not necessarily imply *asymptotic individual calibration*, $P(\mu_1(x) \le y \le \mu_2(x) \mid x) \to 1 - \alpha$. This distinction is now noted via a footnote on page 10 of the revised manuscript.

## 3   Reviewer MNi4

**Comment 1:** Broader and stronger baselines for uncertainty quantification:- The experimental comparison in the neural network setting is currently limited to a relatively narrow set of PI-specific methods. There exist several widely used and competitive uncertainty quantification approaches that are not included, such as Monte Carlo Dropout , deep ensembles, and likelihood-based regression models that learn Gaussian (or mixture) predictive distributions by estimating means and variances. These methods are standard baselines in the uncertainty estimation literature and should be discussed in the Related Work section and, where feasible, included in the experimental comparisons to strengthen the empirical claims.

**Response:** To address this concern we have now substantially revised the *Related Work* section. We have added a detailed discussion of the baseline methods for addressing *aleatoric* and *epistemic* uncertainties. We have also added some new experimental results as suggested by the reviewer which we hope help in strengthening our empirical claims.

Specifically, we have made the following changes.

1. We have added a dedicated discussion of likelihood-based regression models for prediction-interval estimation, with appropriate citations, in the first paragraph of Section 2.1 of the revised manuscript. This clarifies their underlying assumptions and highlights their role in modeling data (aleatoric) uncertainty.

2. Section 2.2 has been added to present a comprehensive discussion of *model (epistemic) uncertainty* in uncertainty quantification, including a systematic review of Bayesian neural networks (MacKay (1992)), MC Dropout (Gal & Ghahramani (2016)), and deep ensemble (Lakshminarayanan et al. (2017)) approaches. These methods are critically examined with respect to their mechanisms for capturing model uncertainty, as well as their practical limitations. We further clarify that, among these alternatives, deep ensembles (Lakshminarayanan et al. (2017)) emerge as a more robust and practically viable option for modeling epistemic uncertainty when compared to Monte Carlo Dropout and fully Bayesian neural networks.

3. To explicitly account for model uncertainty in our experimental study, we have extended the evaluation by incorporating deep ensemble strategies alongside the proposed Tube loss and other existing prediction-interval estimation methods. The revised manuscript now includes **Table 7**, which presents a detailed comparative analysis of deep-ensemble-based PI estimation approaches, including the proposed Tube loss framework.

4. Furthermore, we have incorporated likelihood-based regression prediction-interval baselines, including Mean–Variance Estimation (MVE) neural networks (Nix & Weigend (1994)), the heteroscedastic two-sided PI model (Kendall & Gal (2017)) and Mixture Density Network (MDN) (Bishop (1994)), each combined with deep ensemble strategies. **These methods**

**are employed as competitive baselines and are quantitatively evaluated against the proposed Tube loss framework in Table 7 of the revised manuscript**.

For clarity (**though not included in the revised manuscript**), we note that likelihood-based regression PI models often fail to exhibit consistent performance across diverse datasets, primarily due to their restrictive distributional assumptions. For instance, under uniform noise, the MVE neural network fails to accurately recover the true prediction interval, whereas distribution-free PI models, such as the Tube neural network and Quantile neural network, yield substantially higher-quality PI estimates. This behavior is illustrated in Figure 1 below.

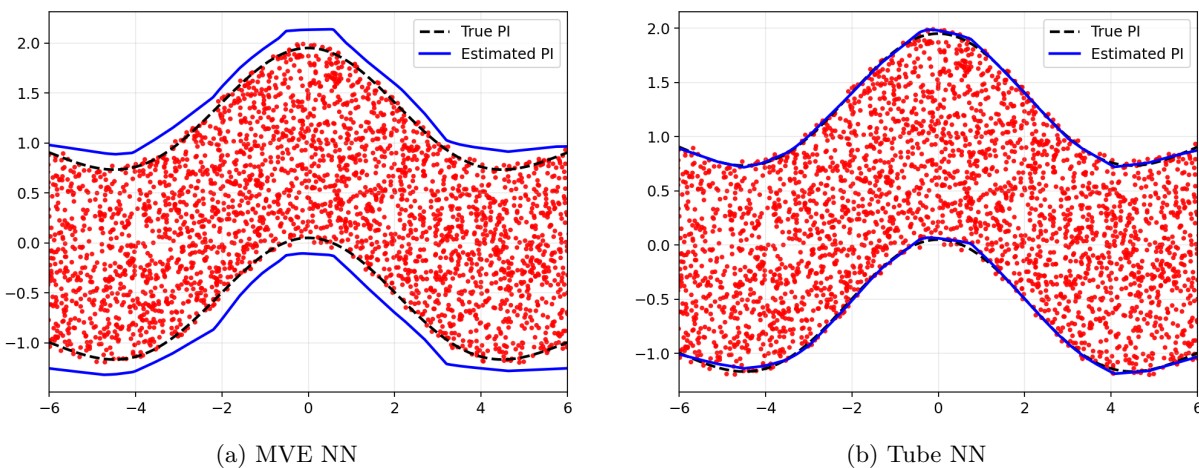

(a) MVE NN            (b) Tube NN

Figure 1: We generate synthetic data points $(x_i, y_i)$ according to the following model: $y_i = \frac{\sin(x_i)}{x_i} + \epsilon_i$, where $\epsilon_i \sim \mathcal{U}(-1, 1)$. The Prediction Intervals (PIs) estimated by the MVE neural network deviate significantly from the true underlying PI, whereas the Tube loss–based neural network produces PI estimates that closely align with the true interval.

**Comment 2:**

While the paper evaluates kernel regression, tabular benchmarks, and time-series forecasting, it would be valuable to demonstrate the effectiveness of the proposed method on regression problems with qualitatively different structures. For example, regression-based tasks in robotics (e.g., behavior cloning or inverse dynamics learning) or computer vision (e.g., monocular depth estimation) are common applications where uncertainty estimation is important and data characteristics differ substantially from time-series settings.

**Response:** In response to this comment, we extend the proposed Tube-loss–based **NN** framework to the *Semantic Textual Similarity (STS)* task, a regression setting that is fundamentally different from the problems considered earlier. Prior work by Wang et al. (2022) highlights the critical role of prediction interval (PI)–based uncertainty quantification in STS, demonstrating that incorporating uncertainty leads to more trustworthy similarity estimates, enhanced robustness, and improved support for downstream decision-making

In pursuance of it, we have added **Subsection 4.6** and **Table 7** to the revised manuscript. In this additional experiment, we consider two widely adopted text representation approaches, namely TF–IDF and Sentence-BERT (SBERT) Reimers & Gurevych (2019). Epistemic (model) uncertainty

is quantified using deep ensembles Lakshminarayanan et al. (2017), while aleatoric (data) uncertainty is modeled via multiple prediction-interval (PI) estimation methods, including Quantile Regression via Gradient Boosting (QRGB), Quantile Regression Neural Networks (QRNN), Heteroscedastic Two-Sided (HTS) PIs Kendall & Gal (2017), Direct Uncertainty Prediction (DUP) Zerva et al. (2022); Wang et al. (2022), Mixture Density Networks (MDN) Bishop (1994), as well as the proposed Tube-loss–based neural network combined with ensembles.

As reported in **Table 7**, across both text representation strategies, the proposed Tube-loss–based ensemble NN consistently achieves superior PI quality and sharper intervals compared to all competing methods. These results empirically validate the effectiveness and generality of the proposed framework on regression tasks whose data characteristics differ substantially from those considered earlier.

| TF–IDF Features | | | SBERT Features | | |
|---|---|---|---|---|---|
| Method | PICP | MPIW | Method | PICP | MPIW |
| Ens–QRGB | 0.9369 | 4.4402 | Ens–QRGB | 0.9007 | 3.9463 |
| Ens–QRNN | 0.8600 | 3.3970 | Ens–HTS | 0.1864 | 0.4052 |
| Ens–HTS | 0.8738 | 3.5094 | Ens–DUP | 0.9101 | 3.4634 |
| Ens–MDN NN | 0.8380 | 3.3230 | Ens–MDN NN | 0.2350 | 0.4960 |
| Ens–Tube NN | **0.9289** | **4.0905** | Ens–Tube NN | **0.9152** | **2.9324** |

Table 2: (Table 7 in main manuscript): Comparative analysis of the Tube NN against existing PI estimation models for the STS task using TF–IDF and SBERT embeddings with target coverage $1 - \alpha = 0.90$. Ens–QRGB: Ensemble Quantile Regression via Gradient Boosting; Ens–QRNN: Ensemble Quantile Regression Neural Network; Ens–HTS: Ensemble Heteroscedastic Two-Sided PI model (Kendall & Gal (2017)); Ens–DUP: Ensemble Direct Uncertainty Prediction (Zerva et al. (2022); Wang et al. (2022)); Ens–MDN: Ensemble Mixture Density Network (Bishop (1994)).

**Comment 3:-** Interpretation of the parameter $r$ in relation to statistical functionals:- The shift parameter plays a central role in the proposed method, yet its statistical interpretation remains somewhat heuristic. It would significantly improve clarity to discuss how different values of $r$ relate to classical point estimators or distributional characteristics, such as the mean, median, or mode of the conditional distribution. Even an intuitive or approximate characterization (e.g., which regimes correspond to more expectation-like or median-like behavior) would help readers better understand and interpret the method.

**Response:** We have added the following discussion in Section 3.3 of the revised manuscript. The optimal choice of the parameter $r$ in the Tube loss is governed by the properties of the conditional distribution $y \mid x$. In particular, setting $r = 0.5$ implicitly assumes symmetry of $y \mid x$, yielding a centered prediction interval (PI) with minimal achievable width. Under this choice, the PI is asymptotically calibrated such that an $\alpha/2$ proportion of samples falls within each of the regions $\Re_1$ and $\Re_4$. Furthermore, in this symmetric setting, the weighted midpoint $r, \mu_1(\boldsymbol{x}) + (1 - r), \mu_2(\boldsymbol{x})$ (shown by the orange line in Figure 3 of the main manuscript) coincides with the conditional median of $y \mid x$.

For asymmetric noise distributions, the optimal choice of $r$ is closely tied to the skewness of the conditional noise. In particular, under right-skewed noise, smaller values of $r$ translate the prediction

interval downward, facilitating the construction of tighter intervals. Conversely, in the presence of left-skewed noise, larger values of $r$ are preferable, yielding improved PI sharpness.

**Comment 4:-**Notation consistency and clarity The paper uses notation inconsistently in several places (e.g., bold versus italic symbols for inputs, reuse of symbols with different meanings, switching between $\alpha$ and other symbols for coverage). A careful pass to standardize notation and clearly define conventions would improve readability and reduce potential confusion for readers.

**Response:** We regret the inconvenience caused by the notation inconsistencies in the original submission. In the revised manuscript, we have taken utmost care to standardize all notations, ensure consistent font usage for scalars and vectors, and adopt $1 - \alpha$ to denote coverage throughout. We have also carefully checked the manuscript for grammatical and typographical errors and made our best efforts to improve overall clarity and readability.

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

|---|---|---|---|---|---|
| **D₁** | Q Ker M | (0.95,0.15) | 0.78 ± 0.017 | 2.088 ± 0.109 | 219.38 |
| | Q Ker M | (0.90,0.10) | 0.78 ± 0.019 | 2.002 ± 0.068 | 221.72 |
| | Q Ker M | (0.85,0.05) | 0.79 ± 0.025 | 2.151 ± 0.078 | 217.38 |
| | T Ker M | (0.6,0) | 0.80 ± 0.012 | 2.165 ± 0.068 | 56.93 |
| | T Ker M | (0.5,0) | 0.80 ± 0.018 | 2.156 ± 0.069 | 61.50 |
| **D₂** | Q Ker M | (0.90,30) | 0.59 ± 0.032 | 4.6609 ± 0.153 | 138.05 |
| | Q Kerl M | (0.95,0.35) | 0.58 ± 0.023 | 5.5369 ± 0.272 | 146.60 |
| | Q Ker M | (0.80,20) | 0.59 ± 0.027 | 3.6196 ± 0.143 | 146.41 |
| | T Ker M | (0.3,0) | 0.60 ± 0.032 | 3.495 ± 0.168 | 39.07 |
| | T Ker M | (0.2,0) | 0.601 ± 0.028 | 3.174 ± 0.165 | 41.55 |

Table 1: Quantile-based Kernel Machine (Q ker M) and Tube loss-based kernel Machine (T ker M) on dataset **D₁** and **D₂**. Q ker M involves parameters (q+1-α,q) and T ker M involves (r, δ).

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

 data point $(x_k, y_k)$ such that $y_k = K(A^T, x_k)\eta + \eta_0$, or $K(A^T, x_k)\beta + \beta_0$, the unique gradient for Tube loss does not exist and any sub-gradient can be considered for computation. The data point $(x_k, y_k)$ lying exactly upon the surface $r((K(A^T, x_k)\beta) + \beta_0) + (1-r)(K(A^T, x_k)\eta + \eta_0)$ can be ignored.

Now, for given data point $(x_i, y_i)$, we consider the width of PI tube $\delta|(K(A^T, x_i, y_i)(\beta-\eta)+(\beta_0-\eta_0)|$ and denote it as $J(\beta, \eta, \beta_0, \eta_0, x_i, y_i)$ and compute

$$\frac{\partial J(\beta, \eta, \beta_0, \eta_0, x_i)}{\partial \beta} = sign((K(A^T, x_i)(\beta-\eta)+(\beta_0-\eta_0))K(A, x_i) \quad \text{if } (K(A^T, x_i)(\beta-\eta)+(\beta_0-\eta_0)) \neq 0$$

$$\frac{\partial J(\eta, \beta, b_1, b_2, x_i)}{\partial \beta_0} = sign((K(A^T, x_i)(\beta-\eta)+(\beta_0-\eta_0)) \quad \text{if } (K(A^T, x_i)(\beta-\eta)+(\beta_0-\eta_0)) \neq 0$$

$$\frac{\partial J(\eta, \beta, b_1, b_2, x_i)}{\partial \eta} = -sign((K(A^T, x_i)(\beta-\eta)+(\beta_0-\eta_0))K(A, x_i) \quad \text{if } (K(A^T, x_i)(\beta-\eta)+(\beta_0-\eta_0)) \neq 0$$

$$\frac{\partial J(\eta, \beta, b_1, b_2, x_i)}{\partial \eta_0} = -sign((K(A^T, x_i)(\beta-\eta)+(\beta_0-\eta_0)) \quad \text{if } (K(A^T, x_i)(\beta-\eta)+(\beta_0-\eta_0)) \neq 0$$

For data point $(x_i, y_i)$ satisfying $(K(A^T, x_i)(\beta-\eta)+(\beta_0-\eta_0) = 0$, a unique gradient of $J(\beta, \eta, \beta_0 - \eta_0, x_i, y_i)$ does not exist, and any subgradient can be considered for computation.

Now, we state gradient descent algorithm for the Tube loss based kernel machine problem.

**Algorithm 2:-**

Input:- Training Set $T = \{(x_i, y_i) : x_i \in \mathbb{R}^p, y_i \in \mathbb{R}, i = 1, 2, ...n\}$, confidence $t \in (0, 1)$ $r, \delta, \gamma$ and $tol$.
Initialize:- $\beta^0, \eta^0 \in \mathbb{R}^n$ and $\beta_0^0, \eta_0^0 \in \mathbb{R}$.

Repeat

$$\beta^{(k+1)} = \beta^{(k)} - \gamma_k\left(\lambda\beta^{(k)} \sum_{i=1}^n \frac{\partial \mathcal{L}_{1-\alpha}^r\left(y_i, \left(K(A^T, x_i)\beta+\beta_0\right), \left(K(A^T, x_i)\eta+\eta_0\right)\right)}{\partial \beta^{(k)}} + \delta \sum_{i=1}^n \frac{\partial J(\beta, \eta, \beta_0, \eta_0, x_i)}{\partial \beta^{(k)}}\right)$$

$$\beta_0^{(k+1)} = \beta_0^{(k)} - \gamma_k\left(\sum_{i=1}^n \frac{\partial \mathcal{L}_{1-\alpha}^r\left(y_i, \left(K(A^T, x_i)\beta+\beta_0\right), \left(K(A^T, x_i)\eta+\eta_0\right)\right)}{\partial \beta_0^{(k)}} + \delta \sum_{i=1}^n \frac{\partial J(\beta, \eta, \beta_0, \eta_0, x_i)}{\partial \beta_0^{(k)}}\right)$$

$$\eta^{(k+1)} = \eta^{(k)} - \gamma_k\left(\lambda\eta^{(k)} \sum_{i=1}^n \frac{\partial \mathcal{L}_{1-\alpha}^r\left(y_i, \left(K(A^T, x_i)\beta+\beta_0\right), \left(K(A^T, x_i)\eta+\eta_0\right)\right)}{\partial \eta^{(k)}} + \delta \sum_{i=1}^n \frac{\partial J(\beta, \eta, \beta_0, \eta_0, x_i)}{\partial \eta^{(k)}}\right)$$

$$\eta_0^{(k+1)} = \eta_0^{(k)} - \gamma_k\left(\sum_{i=1}^n \frac{\partial \mathcal{L}_{1-\alpha}^r\left(y_i, \left(K(A^T, x_i)\beta+\beta_0\right), \left(K(A^T, x_i)\eta+\eta_0\right)\right)}{\partial \eta_0^{(k)}} + \delta \sum_{i=1}^n \frac{\partial J(\beta, \eta, \beta_0, \eta_0, x_i)}{\partial \eta_0^{(k)}}\right)$$

Until $\left\| \begin{bmatrix} \beta^{(k+1)} - \beta^{(k)} \\ \beta_0^{(k+1)} - \beta_0^{(k)} \\ \eta^{(k+1)} - \eta^{(k)} \\ \eta_0^{(k+1)} - \eta_0^{(k)} \end{bmatrix} \right\| \geq tol.$

In our implementation [1], the gradient descent algorithm for the Tube loss kernel machine initially utilizes only the gradient of the Tube loss function. After a certain number of iterations, once we confirm that the upper bound of the PI, $K(A^T x_i)\beta + \beta_0$, has moved above the lower bound, $K(A^T x_i)\eta + \eta_0$, of PI, we incorporate the gradient of the PI tube width into the training of the Tube loss kernel machine.

## C   Additional Experimental Details

**Tuning $r$ parameter:-** Ideally, the tuning of the $r$ parameter in the tube loss function should align with the skewness of the distribution $y|x$. However, real-world datasets often exist in high dimensions. Consequently, for a specific value of $x$, there may be only a few of $y_i$ values in the

---

[1]A MATLAB implementation of gradient descent with Tube loss is provided in supplementary material code.

training dataset, leading to potential inaccuracies in estimating the skewness of the distribution $y|x$. In practical benchmark experiments, we have tuned the $r$ values for the tube loss machine from the set $\{0.1, 0.2, ..., 0.9\}$.

**Tuning $\delta$ parameter:-** Initially, we initialize $\delta$ to zero in the Tube loss-based machine for benchmark dataset experiments and aim to achieve the narrowest PI by adjusting the $r$ parameter, either moving the PI tube upwards or downwards. Once the $r$ parameter is set and if the Tube loss-based machine achieves a coverage higher than the target $t$ on the validation set, we gradually fine-tune the $\delta$ parameter from the set $\{0.001, 0.005, 0.1, 0.15, 0.2\}$ to minimize the tube width while maintaining the desired target coverage $t$.

For the numerical results in Table 2, the tuned parameters for the Tube Loss are summarized in Table 8.

| dataset | r (tube_r) | delta (penalty) | lr | batch | dropout | epochs |
|---|---|---|---|---|---|---|
| boston | 0.5 | 0.03 | 0.05 | 10000 | 0.2 | 400 |
| concrete | 0.25 | 0.05 | 0.015 | 64 | 0.25 | 150 |
| cpu_act | 0.5 | 0.03 | 0.05 | 10000 | 0.2 | 400 |
| energy | 0.9 | 0.01 | 0.05 | 10000 | 0.2 | 400 |
| kin8nm | 0.1 | 0.01 | 0.05 | 10000 | 0.2 | 400 |
| miami | 0.3 | 0.05 | 0.05 | 10000 | 0.2 | 400 |
| naval | 0.3 | 0.05 | 0.05 | 10000 | 0.2 | 400 |
| power | 0.3 | 0.1 | 0.05 | 10000 | 0.2 | 400 |
| protein | 0.1 | 0.01 | 0.05 | 10000 | 0.2 | 400 |
| sulfur | 0.1 | 0.01 | 0.05 | 10000 | 0.2 | 400 |
| wine | 0.5 | 0.05 | 0.05 | 10000 | 0.2 | 400 |
| yacht | 0.5 | 0.03 | 0.01 | 10000 | 0.1 | 400 |

Table 8: Best hyper-parameters for Tube Loss based NN for results generated in Table 2.

| Dataset (Size) | LSTM Structure | Batch Size | | Up,Low | r,$\delta$ | Window Size | Learning Rate | |
|---|---|---|---|---|---|---|---|---|
| | | Q- LSTM | T-LSTM | Q- LSTM | T-LSTM | | Q-LSTM | T-LSTM |
| Sunspots (3265) | [256(0.3) ,128(0.2)] | 128 | 128 | 0.98,0.03 | 0.5,0 | 16 | 0.001 | 0.001 |
| Electric Production (397) | [64] | 32 | 16 | 0.98,0.03 | 0.5,0.01 | 12 | 0.01 | 0.001 |
| Daily Female Birth (365) | [100] | 64 | 128 | 0.98,0.03 | 0.5,0.01 | 12 | 0.01 | 0.005 |
| SWH (2170) | [128(0.4),64(0.3),32(0.2)] | 300 | 300 | 0.98, 0.03 | 0.5, 0.01 | 100 | 0.001 | 0.001 |
| Temperature (3651) | [16,8] | 300 | 300 | 0.98, 0.03 | 0.5, 0.01 | 100 | 0.001 | 0.0001 |
| Beer Production (464) | [64,32] | 64 | 64 | 0.98, 0.03 | 0.1, 0.01 | 12 | 0.001 | 0.0001 |

Table 9: Tuned parameter values for Q-LSTM and T-LSTM model for considered benchmark datasets in Table 3. The LSTM architecture [128(0.4),64(0.3),32(0.2)] means that three hidden layers with neurons 128, 64 and 32 respectively and each hidden layer is followed by a drop out layer which are 0.4,0.3 and 0.2 respectively.

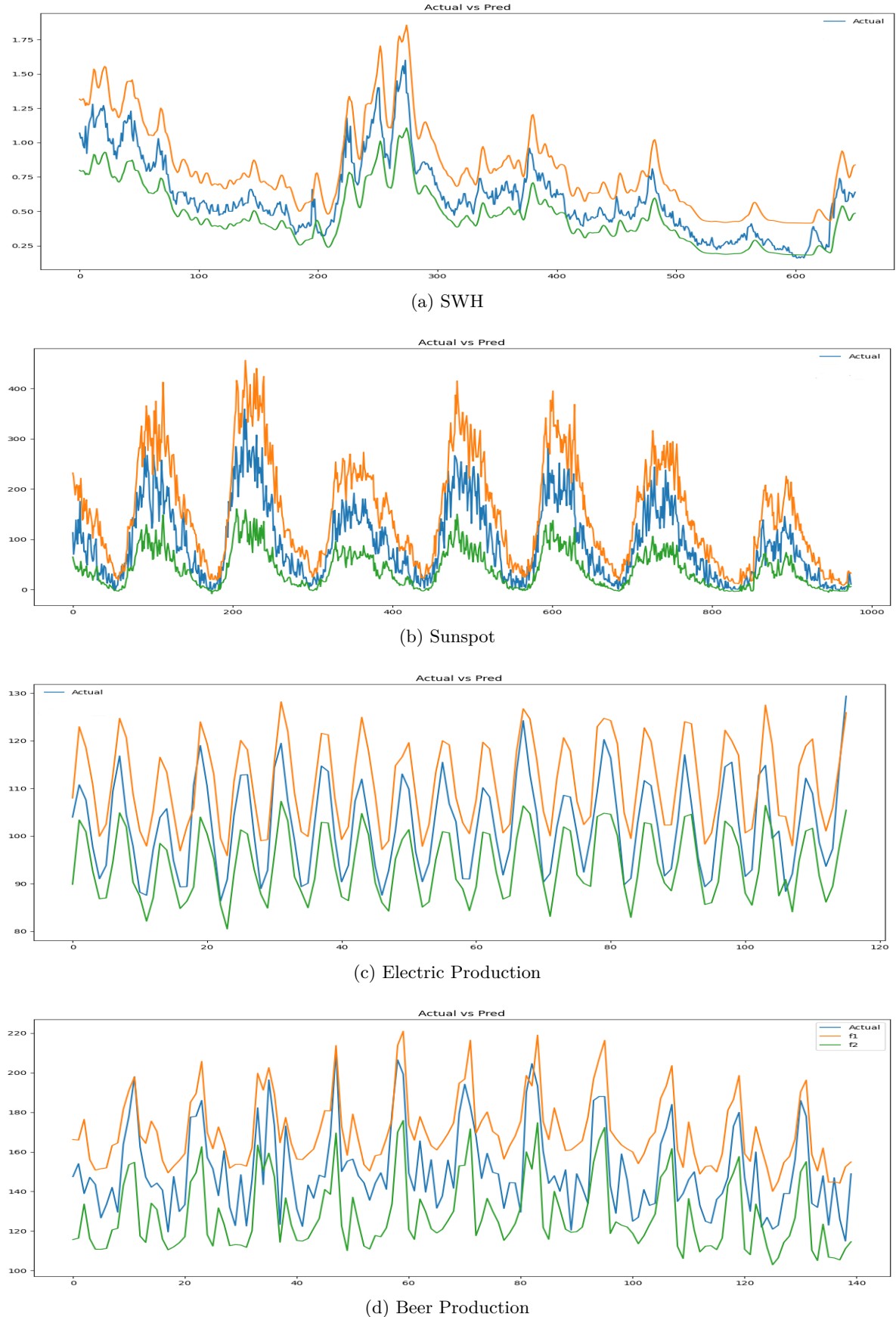

Figure 11: Probabilistic forecast of LSTM with proposed Tube loss function.