# OpenReview forum: "Tube Loss: A Novel Loss Function for Prediction Interval Estimation"
_TMLR — Accepted by TMLR_

### Review · Reviewer_MNi4 · 2025-12-12

**Summary Of Contributions:**

This paper introduces Tube Loss, a new loss function for directly learning prediction intervals in regression models. The method estimates the lower and upper bounds simultaneously in a single optimization problem and can be trained efficiently using gradient-based methods. A tunable parameter allows the interval to shift along the response distribution, which is particularly useful for asymmetric or skewed data. The authors provide theoretical results showing that the learned intervals asymptotically attain the target coverage level. Empirical results across kernel methods, neural networks, and probabilistic forecasting demonstrate improved interval quality compared to existing approaches.

**Audience:**

Yes

**Audience Explanation:**

The paper addresses prediction interval estimation, which is a core topic in uncertainty quantification and regression modeling. The proposed loss-based approach offers a practical alternative to quantile-based and ensemble-based methods and may be of interest to researchers working on uncertainty-aware learning, probabilistic forecasting, and applied machine learning.

**Broader Impact Concerns:**

The paper focuses on methodological advances in prediction interval estimation and does not raise immediate ethical concerns.

**Claims And Evidence:**

Yes

**Claims Explanation:**

The submission provides both theoretical and empirical evidence to support its main claims. The proposed loss function is clearly defined, its asymptotic coverage properties are analyzed under explicit assumptions, and the empirical results across multiple model classes and datasets are generally consistent with the stated objectives. While the guarantees are asymptotic rather than finite-sample, the claims are framed accordingly and do not overreach the presented analysis.

**Requested Changes:**

Major points

1. Broader and stronger baselines for uncertainty quantification

The experimental comparison in the neural network setting is currently limited to a relatively narrow set of PI-specific methods. There exist several widely used and competitive uncertainty quantification approaches that are not included, such as Monte Carlo Dropout [1], deep ensembles, and likelihood-based regression models that learn Gaussian (or mixture) predictive distributions by estimating means and variances [2]. These methods are standard baselines in the uncertainty estimation literature and should be discussed in the Related Work section and, where feasible, included in the experimental comparisons to strengthen the empirical claims.

2. Evaluation on non–time-series regression tasks beyond standard benchmarks

While the paper evaluates kernel regression, tabular benchmarks, and time-series forecasting, it would be valuable to demonstrate the effectiveness of the proposed method on regression problems with qualitatively different structures. For example, regression-based tasks in robotics (e.g., behavior cloning or inverse dynamics learning) or computer vision (e.g., monocular depth estimation) are common applications where uncertainty estimation is important and data characteristics differ substantially from time-series settings.

3. Interpretation of the parameter $r$ in relation to statistical functionals

The shift parameter $r$ plays a central role in the proposed method, yet its statistical interpretation remains somewhat heuristic. It would significantly improve clarity to discuss how different values of $r$ relate to classical point estimators or distributional characteristics, such as the mean, median, or mode of the conditional distribution. Even an intuitive or approximate characterization (e.g., which regimes of $r$ correspond to more expectation-like or median-like behavior) would help readers better understand and interpret the method.

Minor

1. Notation consistency and clarity
The paper uses notation inconsistently in several places (e.g., bold versus italic symbols for inputs, reuse of symbols with different meanings, switching between $\alpha$ and other symbols for coverage). A careful pass to standardize notation and clearly define conventions would improve readability and reduce potential confusion for readers.

[1] Gal, Yarin, and Zoubin Ghahramani. "Dropout as a bayesian approximation: Representing model uncertainty in deep learning." international conference on machine learning. PMLR, 2016.
[2] Kendall, Alex, and Yarin Gal. "What uncertainties do we need in bayesian deep learning for computer vision?." Advances in neural information processing systems 30 (2017).

---

> ### Author Response · Authors · 2026-02-21
> **Response to Reviewer**
>
> We gratefully acknowledge the reviewers’ thoughtful and constructive comments and believe that careful consideration of these suggestions has meaningfully strengthened the quality and clarity of our work. We have attached a response letter as supplementary material in our submission that presents our point-by-point responses to the reviewers’ observations and recommendations. A copy of the revised manuscript is attached with this response letter, where all changes have been highlighted in blue for ease of reference.
>
> We have significantly revised our manuscript in the light of  reviewer's valuable comments and incorporated reviewer suggestions in the revised manuscript.  Please see the response letter in supplementary material.

---

### Review · Reviewer_QADY · 2025-12-21

**Summary Of Contributions:**

The authors proposed a new loss function, called the tube loss, for directly learning prediction intervals. The tube loss can be optimized using gradient-based methods, and is linear in the difference between the observation and the upper/lower bound. The resulting interval from the tube loss does not encounter crossing issues and can be adjusted to the density of the observed responses in the training set. They also allow users to tune the interval according to their preference for coverage or the length of the interval. The authors also theoretically proved that the resulting interval will asymptotically attain the nominal coverage. Empirical experiments are conducted to support their findings.

**Additional Comments:**

NA.

**Audience:**

Yes

**Audience Explanation:**

Distribution-free prediction intervals are of interest to practitioners in both statistics and machine learning. Methods based on conformal inference or based on quantiles are quite popular nowadays. The tube loss proposed in this paper seems to have several interesting features to serve as a promising direction. In summary, I believe there should be individuals in TMLR's audience interested in this paper.

**Broader Impact Concerns:**

NA.

**Claims And Evidence:**

Yes

**Claims Explanation:**

I chose "yes", but I want to emphasize that the paper has too many typos and notation inconsistencies (see details later) to judge it effectively. I firmly believe that the paper should be rewritten to be better assessed. The empirical results in this paper are promising.

**Requested Changes:**

The paper has some merit. However, in my own opinion, it requires significant changes.

(1) There are many grammatical errors. For example, on page 6, there is one sentence, "... some good theoretical properties, such as, it attains coverage, and it is unbiased and consistent", containing multiple grammatical issues. It is the authors' responsibility to make a significant effort to improve the writing.

(2) There are also many notation inconsistencies. For example, again on page 6, equation (12), there is no input for $\mathcal{L}_{1 - \alpha}^{RQR}$.

I also highly recommend that the authors make some effort to make the mathematical notation used in the paper aesthetically appealing. For instance, $\mathcal{L}_{LUBE}$ looks quite annoying.

(3) In Figure 1, what is $t$? A visual comparison between the tube loss and the RQR loss is more recommended than displaying the RQR loss in almost the end of the paper.

(4) The theoretical results should be stated with more care. For instance, in both Lemma 1 and Proposition 1, some of the notations are not defined in the paper, such as $n_{1}, \cdots, n_{4}$.

(5) In section 3.3, what is $\mathcal{M}$? Should it be $\mathcal{L}$ instead?

(6) Numerical comparisons should be more comprehensive. In the current version, the tube loss-based interval is only compared against one or two alternative methods.

(7) The theoretical results are somewhat preliminary. This is not a very serious issue for TMLR. But I would like to see that the authors could discuss more caveats about the theoretical results. For instance, when gradient-based methods are used, even though the conditional density satisfies certain smoothness conditions, it is not necessarily the case that the eventual output converges to the true quantile regression functions, and thus the cited results of Takeuchi et al. may not be directly applicable. Proposition 1 should therefore be stated with more caution.

---

> ### Author Response · Authors · 2026-02-21
> **Response to Reviewer**
>
> We gratefully acknowledge the reviewers’ thoughtful and constructive comments and believe that careful consideration of these suggestions has meaningfully strengthened the quality and clarity of our work. We have attached a response letter as supplementary material in our submission that presents our point-by-point responses to the reviewers’ observations and recommendations. A copy of the revised manuscript is attached with this response letter, where all changes have been highlighted in blue for ease of reference.
>
>  We have significantly revised our manuscript in the light of  reviewer's valuable comments and incorporated reviewer suggestions in the revised manuscript.  Please see the response letter in supplementary material.

---

### Review · Reviewer_RuFc · 2026-01-25

**Summary Of Contributions:**

### Summary:
This paper proposes a new loss function that simultaneously estimates prediction interval bounds. It introduces a tuning parameter to shift intervals for skewed distributions and uses linear penalties for robustness against outliers. Experiments cover kernel machines, NNs, and LSTMs.

### Weakness:
- The novelty appears increamental. The motivation is to get both bounds from one optimization, but in practice training two separate quantile models is already standard and computationally efficient.
- In a related spirit, the proposed method needs a grid search for the hyperparameter $r$ through multiple runs, so it is also computationally expensive and not a single optimization as claimed.
- Conformal prediction is a state-of-the-art baseline in uncertainty quantification, while the authors excludes them due to reasons like different framework or computation cost.

**Additional Comments:**

None

**Audience:**

Yes

**Audience Explanation:**

Uncertainty quantification of machine-learning predictions is an interesting topic, especially for safety-critical applications.

**Claims And Evidence:**

No

**Claims Explanation:**

- The claim of "single optimization" is misleading. The proposed method requires a grid search for the hyperparameter $r$ through runs, which makes the total training cost higher than standard quantile regression.
- The authors claimed that conformal prediction requires retraining from scratch. However, standard methods are efficient post-hoc steps that do not require retraining.
- Computation-time reporting is not comparable as presented. For a fair comparison, the paper should report end-to-end computation time including hyperparameter tuning, especially since the proposed method appears to require extensive tuning across multiple runs.

**Requested Changes:**

- Compare with the state-of-the-art conformal baseline.
- Report computation time including tuning for fair comparison. The proposed method requires tuning across multiple runs.
- Provide stronger theoretical or empirical justification for the proposed method.

---

> ### Author Response · Authors · 2026-02-21
> **Response to Reviewer**
>
> We gratefully acknowledge the reviewers’ thoughtful and constructive comments and believe that careful consideration of these suggestions has meaningfully strengthened the quality and clarity of our work. We have attached a response letter as supplementary material in our submission that presents our point-by-point responses to the reviewers’ observations and recommendations. A copy of the revised manuscript is attached with this response letter, where all changes have been highlighted in blue for ease of reference.
>
> We have significantly revised our manuscript in the light of  reviewer's valuable comments and incorporated reviewer suggestions in the revised manuscript.  Please see the response letter in supplementary material.

---

### Decision · Action_Editor_8v4p · 2026-03-15

**Recommendation:** Accept as is

**Audience:**

Yes

**Audience Explanation:**

The audience of this paper could be broad: this is a method development for uncertainty quantification with predictions, and many areas of ML and statistics would find it interesting.

**Claims And Evidence:**

Yes

**Claims Explanation:**

The authors proposed a novel loss function, termed Tube Loss, for the simultaneous estimation of both lower and upper bounds of a predictive interval (PI). Both theoretical and empirical properties of this new loss function have been studied. The authors also compared with some other loss functions under different scenarios.